# FLEXIBLE ACTIVE LEARNING OF PDE TRAJECTORIES

## ABSTRACT

Accurately solving partial differential equations (PDEs) is critical for understanding complex scientific and engineering phenomena, yet traditional numerical solvers are computationally expensive. Surrogate models offer a more efficient alternative, but their development is hindered by the cost of generating sufficient training data from numerical solvers. In this paper, we present a novel framework for active learning (AL) in PDE surrogate modeling that reduces this cost. Unlike the existing AL methods for PDEs that always acquire entire PDE trajectories, our approach strategically generates only the most important time steps with the numerical solver, while employing the surrogate model to approximate the remaining steps. This dramatically reduces the cost incurred by each trajectory and thus allows the active learning algorithm to try out a more diverse set of trajectories given the same budget. To accommodate this novel framework, we develop an acquisition function that estimates the utility of a set of time steps by approximating its resulting variance reduction. We demonstrate the effectiveness of our method on several benchmark PDEs, including the Heat equation, Korteweg–De Vries equation, Kuramoto–Sivashinsky equation, and the incompressible Navier-Stokes equation. Extensive experiments validate that our approach outperforms existing methods, offering a cost-efficient solution to surrogate modeling for PDEs.

## 1 INTRODUCTION

In many scientific and engineering applications, accurately solving partial differential equations (PDEs) in the form of trajectories of states evolving over time is essential for understanding complex phenomena (Holton & Hakim, 2013; Atkins et al., 2023; Murray, 2007; Wilmott et al., 1995). The traditional approach involves running numerical solvers, which provide accurate solutions but are computationally costly, taking several hours, days or even weeks to run depending on the complexity of the problem (Cleaver et al., 2016; Cowan et al., 2001). As a result, there is significant interest in developing surrogate models (Greydanus et al., 2019; Bar-Sinai et al., 2019; Sanchez-Gonzalez et al., 2020; Li et al., 2020; Brandstetter et al., 2022b; Lippe et al., 2024) that can approximate the solutions more efficiently. Surrogate models are obtained by solving regression tasks on some "ground truth" data. The ground truth data for PDEs are generated by numerical solvers, which are costly compared to those of standard regression problems. As a result, the expense of data acquisition presents a major bottleneck in the development of surrogate models for PDEs.

Active Learning (AL, Chernoff, 1959; MacKay, 1992; Settles, 2009) can address this challenge by adaptively acquiring the most informative inputs, effectively reducing the amount of ground-truth data required to obtain a high-quality surrogate model. However, there is a general lack of research in AL for regression tasks (Wu, 2018; Holzmüller et al., 2023), let alone PDEs. Existing studies on AL for PDEs have predominantly dealt with univariate outputs such as energy (Pestourie et al., 2020; Pickering et al., 2022), or predictions at a single, fixed time point (Bajracharya et al., 2024; Wu et al., 2023b). To our surprise, the only work directly addressing AL for prediction of trajectories is that of Musekamp et al. (2024). In this work, the surrogate model is set as an autoregressive model that predicts the evolved state of a PDE at time $t + \Delta t$ given a state at an arbitrary time point $t$, and is trained on data acquired by existing regression-based AL methods (Holzmüller et al., 2023). Specifically, at each round of acquisition, the AL method chooses initial conditions from which *entire* trajectories are acquired. However, we argue that querying all the states in a trajectory is not sample-efficient, especially for autoregressive surrogate models.

**Figure 1:** Conceptual illustration of our framework for data acquisition. Each dot represents a PDE state, and a path connecting two dots represents a time step of a simulation. Black solid lines are obtained with a numerical solver and red dotted lines with a surrogate model.

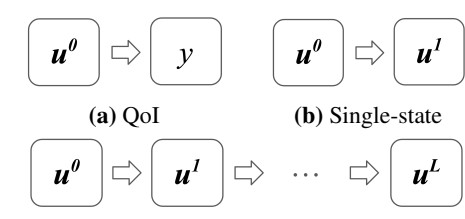

**(a)** QoI      **(b)** Single-state

**(c)** Autoregressive

**Figure 2:** Task settings assumed by previous works in active learning of PDEs.

Acquiring entire trajectories is inefficient mainly for two reasons. First, states within a trajectory are often strongly correlated, undermining their diversity or the joint information gain (Houlsby et al., 2011; Kirsch et al., 2019). We validate this assertion in Appendix C.1 through principal component analysis. Secondly, even if they are not strongly correlated, it can be the case that only certain time steps of a trajectory are the most informative due to the dynamics of the PDE. In both cases, noting that the main cost is in running the numerical solver, it would be ideal to selectively acquire only the most important time steps with the numerical solver, for a fraction of the cost of acquiring the entire trajectory. However, this is usually impossible without querying all the time steps that come earlier.

In this paper, we propose a novel, flexible framework for data acquisition that circumvents the constraint of having to query all time steps in a trajectory, along with an AL strategy that leverages this flexibility. Our method combines both a numerical solver and a surrogate model to acquire data along a trajectory with reduced cost. Specifically, it selects which time steps along a trajectory to query from the solver, while using the surrogate model to approximate the remaining steps. We also develop a novel acquisition function that guides our AL strategy in choosing which time steps to query to the numerical solver in each trajectory.

Overall, our framework, equipped with the novel AL strategy, significantly improves surrogate model performance over previous methods. We validate our approach through extensive experiments on benchmarks, including the Heat equation, Korteweg–De Vries equation, Kuramoto–Sivashinsky equation, and the Navier-Stokes equation. Additionally, we analyze the behavior of our AL method, providing insights into the factors that contribute to its effectiveness.

## 2 BACKGROUND

### 2.1 PRELIMINARIES

We consider PDEs with one time dimension $t \in [0, T]$ and possibly multiple spatial dimensions $\boldsymbol{x} = [x_1, x_2, \ldots, x_D] \in \mathbb{X}$ where $\mathbb{X}$ is the spatial domain such as the unit interval. These can be written in the form

$$\partial_t \boldsymbol{u} = F(t, \boldsymbol{x}, \boldsymbol{u}, \partial_{\boldsymbol{x}} \boldsymbol{u}, \partial_{\boldsymbol{xx}} \boldsymbol{u}, \ldots), \tag{1}$$

where $\boldsymbol{u} : [0, T] \times \mathbb{X} \to \mathbb{R}^n$ is a solution to the PDE. We are also given a specific boundary condition and a fixed time interval $\Delta t$. If the PDE is well-posed (Evans, 1988), there exists, for each $t_0 \in \mathbb{R}$, an evolution operator $G_{t_0}$ which maps an initial condition $\boldsymbol{u}^0 := \boldsymbol{u}(t_0, \cdot)$ to the solution $\boldsymbol{u}^1 := \boldsymbol{u}(t_0 + \Delta t, \cdot)$. For simplicity, we only consider time-independent PDEs, for which the evolution operator $G_t$ is the same for all $t$, say $G$. Iterating over $G$ multiple times, we can obtain a trajectory $\left(\boldsymbol{u}^i\right)_{i=1}^{L}$ of length $L$, where $\boldsymbol{u}^i := G^{(i)}[\boldsymbol{u}^0]$ with $G^{(i)}$ being the $i$-th iterate of $G$. Although a numerical solver $G_{\text{solver}}$ is only an approximation to $G$, we shall not distinguish between the two for the remainder of this paper.

There are three primary tasks in active learning for PDEs, each depending on the type of surrogate model being trained. The first task, *univariate Quantity of Interest (QoI) prediction*, focuses on

learning a model to directly predict a scalar QoI, denoted as $y$, from an initial condition $\boldsymbol{u}^0$. The second task, *single-state prediction*, involves learning a model to predict a single state transition from $\boldsymbol{u}^0$ to $\boldsymbol{u}^1$ over a fixed time interval $\Delta t$. The third task, *autoregressive trajectory prediction*, aims to approximate the ground truth evolution operator $G$ using a surrogate model to predict the entire time evolution of the states. Fig. 2 provides a visual comparison of the three tasks. In this paper, we focus on the autoregressive trajectory prediction task.

We train a neural surrogate model $\hat{G}$ with input-output pairs $(\boldsymbol{u}^{i-1}, \boldsymbol{u}^i)$ from the numerical solver $G$. Active learning builds a high quality training dataset by adaptively selecting informative inputs to be fed into the solver $G$. Prior work (Musekamp et al., 2024) operates on the the framework where initial conditions $\boldsymbol{u}^0$ are selected from a pool $\mathcal{P}$, from which full trajectories of length $L$ are obtained. For instance, Query-by-Committee (**QbC**, Seung et al., 1992) queries initial conditions $\boldsymbol{u}^0$ that maximize the predictive uncertainty estimated from a committee of $M$ models,

$$a_{\text{QbC}}(\boldsymbol{u}^0) = \frac{1}{M} \sum_{m=1}^{M} \sum_{i=1}^{L} \|\hat{\boldsymbol{u}}_m^i - \bar{\hat{\boldsymbol{u}}}^i\|_2^2 \tag{2}$$

where $\hat{\boldsymbol{u}}_m^i$ is the prediction of the $i^{\text{th}}$ state from the $m^{\text{th}}$ surrogate model in the committee and $\bar{\hat{\boldsymbol{u}}}^i := \frac{1}{M} \sum_{m=1}^{M} \hat{\boldsymbol{u}}^i$ is the mean prediction from the committee.

## 2.2 PROBLEM SETTING

Our ultimate objective is to obtain a surrogate model $\hat{G}$ that approximates the expensive numerical solver $G$ with low error

$$\frac{1}{N_{\text{test}}} \sum_{j=1}^{N_{\text{test}}} \text{err}\left( (G^{(i)}[\boldsymbol{u}_j^0])_{i=1}^L, (\hat{G}^{(i)}[\boldsymbol{u}_j^0])_{i=1}^L \right) \tag{3}$$

where $\text{err}(\cdot, \cdot)$ is an error metric. Obtaining the surrogate model requires sampling training data from the numerical solver, which incurs a nontrivial cost. AL aims to improve sample efficiency by sampling only the most important data. In particular, AL utilizes the current surrogate model $\hat{G}$, or a committee of surrogate models $\{\hat{G}_m\}_{m=1}^M$, to inform its choice. After acquiring the data chosen by AL, we retrain the surrogate $\hat{G}$ with the expanded training dataset.

We assume that there exists a pool $\mathcal{P}$ of initial conditions $\boldsymbol{u}^0$. At each round of AL, we train a committee of $M$ surrogate models $\{\hat{G}_m\}_{m=1}^M$ with the training dataset collected from $G$ up to that round. We then use this committee to select a batch of inputs to be queried to the solver $G$ and add them to the training dataset. The cost at each round, defined as the number of inputs queried to the numerical solver $G$, is limited to a certain budget $B$. Our aim is to achieve low errors at each round, so an AL strategy would ideally acquire data with cost as close to or equal to the budget (Li et al., 2022a).

## 3 FLEXIBLE ACTIVE LEARNING FOR PDEs

### 3.1 FRAMEWORK OF DATA ACQUISITION

We present our method, FLEXAL, which operates under a framework of data acquisition that is much more sample efficient than previous works. Algorithm 1 provides an overview of our framework. We start with a surrogate model $\hat{G}$, or a committee of surrogate models $\{\hat{G}_m\}_{m=1}^M$, trained with the initial dataset $\mathcal{D}$. At every round of AL, we choose an initial condition $\boldsymbol{u}^0$ from the pool $\mathcal{P}$, similar to the existing AL methods for PDE trajectories. However, while existing methods acquire the entire trajectory starting from the chosen initial condition $\boldsymbol{u}^0$ (Musekamp et al., 2024), our method acquires a partial trajectory. Specifically, we select a subset of time steps to simulate from $\boldsymbol{u}^0$, rather than acquiring the full trajectory. The rationale behind this approach is that, given a fixed budget, acquiring as many trajectories as possible—albeit partially—from different initial conditions is often more beneficial than fully acquiring fewer trajectories. This strategy enables more efficient exploration of the data space and improves the overall sample efficiency of the framework.

---

**Algorithm 1** Overview of Flexible Active Learning (FLEXAL)

---

**Require:** Pool $\mathcal{P}$ of initial conditions, budget $B$ per round, number of rounds $R$, numerical solver $G$, trajectory length $L$, initial training dataset $\mathcal{D}$
**Ensure:** Trained surrogate model $\hat{G}$

1: Train $\hat{G}$ on $\mathcal{D}$
2: **for** round $= 1$ to $R$ **do**
3:     $\text{cost} \leftarrow 0$
4:     **while** $\text{cost} < B$ **do**
5:         Choose initial condition $\boldsymbol{u}^0$ from $\mathcal{P}$                    ▷ Section 3.3
6:         $\mathcal{P} \leftarrow \mathcal{P} \setminus \{\boldsymbol{u}^0\}$
7:         Choose sampling pattern $S = (b_1, \ldots, b_L)$              ▷ Sections 3.2 and 3.3
8:         $\hat{\boldsymbol{u}}^0 \leftarrow \boldsymbol{u}^0$
9:         **for** $i = 1$ to $L$ **do**
10:             $\hat{\boldsymbol{u}}^i \leftarrow \begin{cases} G[\hat{\boldsymbol{u}}^{i-1}] & \text{if } b_i = \text{true} \\ \hat{G}[\hat{\boldsymbol{u}}^{i-1}] & \text{if } b_i = \text{false} \end{cases}$
11:             **if** $b_i = \text{true}$ **then**
12:                 $\mathcal{D} \leftarrow \mathcal{D} \cup \{(\hat{\boldsymbol{u}}^{i-1}, \hat{\boldsymbol{u}}^i)\}$
13:                 $\text{cost} \leftarrow \text{cost} + 1$
14:             **end if**
15:         **end for**
16:     **end while**
17:     Train $\hat{G}$ on $\mathcal{D}$
18: **end for**

---

More specifically, for a given initial condition $\boldsymbol{u}^0$, we define a boolean sequence of length $L$, $S = (b_1, \ldots, b_L)$, which we refer to as the *sampling pattern*. For example, $S$ could be $(\text{true}, \text{false}, \ldots, \text{true})$. The sampling pattern specifies that data will be acquired only at time steps corresponding to true values while skipping those marked false.

After selecting the sampling pattern $S$, the next step is to acquire the PDE trajectory. While acquiring a full trajectory is straightforward using a numerical solver $G$, obtaining a partial trajectory corresponding to $S$ can be tricky. We want to run the solver $G$ only for the time steps specified by $S$ (those with true patterns), but the solver requires the skipped time steps (those with false patterns) as intermediate inputs. If we just run the solver for all time steps for this reason, we wouldn't be saving any cost. To address this, we use a simple heuristic: for the skipped time steps, we replace the simulation with predictions from the *surrogate model* (we use the average surrogate $\hat{G} = \frac{1}{M} \sum_{m=1}^{M} \hat{G}_m$ when we have a committee). That is, starting with $\hat{\boldsymbol{u}}^0 = \boldsymbol{u}^0$, we iterate over $1 \leq i \leq L$:

$$\hat{\boldsymbol{u}}^i = \begin{cases} G[\hat{\boldsymbol{u}}^{i-1}] & \text{if } b_i = \text{true} \\ \hat{G}[\hat{\boldsymbol{u}}^{i-1}] & \text{if } b_i = \text{false}. \end{cases} \tag{4}$$

We add to our dataset $\mathcal{D}$ only the input-output pairs obtained with the solver $G$, namely $(\hat{\boldsymbol{u}}^{i-1}, \hat{\boldsymbol{u}}^i)$ with $b_i = \text{true}$.

In comparison to full trajectory acquisition, which requires $L$ executions of the numerical solver, our strategy invokes the numerical solver $\|S\| := \sum_{i=1}^{L} \mathbb{1}[b_i = \text{true}]$ times and utilizes the surrogate model $L - \|S\|$ times. Since the surrogate model is significantly cheaper to evaluate than the numerical solver, this approach substantially reduces the cost of acquisition, enabling us to explore more initial conditions within the same budget. In fact, as discussed in Section 2.2, we define the acquisition cost precisely as $\|S\|$. We repeat expanding our training dataset with new initial conditions and sampling patterns until the cost incurred in the current round reaches a budget $B$. At the end of each round, we retrain the surrogate $\hat{G}$ with the expanded training dataset $\mathcal{D}$.

Previous methods listed in Musekamp et al. (2024) can be considered a special case of ours where the sampling pattern $S$ is always full of true entries. Our framework is therefore a strict generalization of previous works. In the remainder of this section, we describe how FLEXAL adaptively chooses initial conditions $\boldsymbol{u}^0$ and sampling patterns $S$.

## 3.2 ACQUISITION FUNCTION

To adaptively select the sampling pattern $S$ with the initial condition $\boldsymbol{u}^0$, we propose a novel acquisition function $a(\boldsymbol{u}^0, S)$ that assesses the utility of $S$. Given a committee $\{\hat{G}_m\}_{m=1}^M$, consider $(\hat{G}_a, \hat{G}_b)$ for some $a, b \in [M] := \{1, \ldots, M\}$ with $a \neq b$. We define the utility of the sampling pattern $S$ for the pair $(\hat{G}_a, \hat{G}_b)$ as the resulting *variance reduction* in the pair's rolled-out trajectories. Specifically, let $\hat{\boldsymbol{u}}_a$ and $\hat{\boldsymbol{u}}_b$ be the trajectories estimated by $\hat{G}_a$ and $\hat{G}_b$, starting from $\boldsymbol{u}^0$. Next, we obtain a rollout using Eq. 4 with our sampling pattern $S$ and surrogate model $\hat{G}_b$, where $\hat{G}_a$ serves as a stand-in for the ground-truth solver $G$. We denote the resulting trajectory as $\hat{\boldsymbol{u}}_{b,S,a}$. The variance reduction is defined as

$$R(a, b, S) := \sum_{i=1}^{L} \left( \|\hat{\boldsymbol{u}}_a^i - \hat{\boldsymbol{u}}_b^i\|^2 - \|\hat{\boldsymbol{u}}_a^i - \hat{\boldsymbol{u}}_{b,S,a}^i\|^2 \right). \tag{5}$$

The sampling pattern $S$ that maximizes $R(a, b, S)$ is the one where the current models $\hat{G}_a$ and $\hat{G}_b$ disagree the most, and acquiring data from $S$ effectively reduces this discrepancy. Our acquisition function is defined as the average variance reduction between all the distinct pairs in the committee:

$$a(\boldsymbol{u}^0, S) = \frac{1}{M(M-1)} \sum_{a,b \in [M], a \neq b} R(a, b, S). \tag{6}$$

We observe that our acquisition function simplifies to QbC in Eq. 2 when $S$ acquires all the time steps, differing only by a constant factor of two. This occurs because, in that case, $\hat{\boldsymbol{u}}_{b,S,a} = \hat{\boldsymbol{u}}_a$, which makes the second term in the summand of Eq. 2 vanish. Consequently, we can interpret our acquisition function as a generalization of QbC that accommodates for the selection of time steps.

As an additional sanity check, consider the scenario where $S$ does not sample any time steps. In this situation, $\hat{\boldsymbol{u}}_{b,S,a} = \hat{\boldsymbol{u}}_b$, leading the two terms in the summand to cancel each other out, resulting in zero variance reduction. Since acquiring no data should yield zero utility, we confirm that our acquisition function behaves as expected in this limiting case. Appendix D.1 further details the precise motivation behind the design of our acquisition function.

## 3.3 BATCH ACQUISITION ALGORITHM

With the acquisition function defined above, we present a batch acquisition algorithm given a pool $\mathcal{P}$ of initial conditions . We define the cost of a batch $\{(\boldsymbol{u}_j^0, S_j)\}_{j=1}^N$ as the total number of queries to the solver, $\sum_{j=1}^N \|S_j\|$. A standard objective is to maximize $\sum_{j=1}^N a(\boldsymbol{u}_j^0, S_j)$ under a budget constraint $\sum_j \|S_j\| \leq B$, to which there is a known approximate solution (Salkin & De Kluyver, 1975) that greedily maximizes the cost-weighted acquisition function $a^*(\boldsymbol{u}^0, S) = a(\boldsymbol{u}^0, S)/\|S\|$ until the total cost exceeds the budget $B$. However, this method faces two problems. First, it's questionable whether the sum $\sum_{j=1}^N a(\boldsymbol{u}_j^0, S_j)$ of individual acquisition values is actually a good representative for the utility of a batch. In fact, numerous works report that picking instances that maximize individual acquisition values can severely underperform compared to methods that take into account the interactions between those instances (Kirsch et al., 2019; Ash et al., 2019). The problem is chiefly attributed to the lack of diversity and representativeness (Wu, 2018) caused by oversampling of small, high value regions (Smith et al., 2023). Secondly, we are actually searching over the *product* pool of the sampling pattern $S$ and the pool $\mathcal{P}$ of initial conditions, whose size is on the order of $O(2^L|\mathcal{P}|)$. Both terms impose significant computational burden on optimizing the cost-weighted objective $a^*(\boldsymbol{u}^0, S)$.

We therefore propose FLEXAL as an add-on to existing AL methods that acquire full trajectories. Specifically, a full-trajectory AL method $\mathcal{A}$, which we call a *base* method, first selects an initial condition $\boldsymbol{u}^0$. Musekamp et al. (2024) introduces several possibilities for such a method, including **QbC** (Seung et al., 1992), Largest Cluster Maximum Distance (**LCMD**, Holzmüller et al., 2023), **Core-Set** (Sener & Savarese, 2017), and stochastic batch active learning (**SBAL**, Kirsch et al., 2023). We then optimize the cost-weighted acquisition function $a(\boldsymbol{u}^0, S)$ over the sampling pattern $S$ while holding $\boldsymbol{u}^0$ fixed, and add the pair $(\boldsymbol{u}^0, S)$ to the current batch. We iterate this two-stage process until the cost of the batch reaches our budget limit. Additionally, if the cost ever exceeds

the budget after adding a pair, we truncate the sampling pattern so that the cost is exactly equal to the budget. By using FLEXAL as an add-on, the diversity and representativeness promoted by base AL methods (Holzmüller et al., 2023; Kirsch et al., 2023; Musekamp et al., 2024) are upheld, and the size of the optimization space for FLEXAL is reduced to $O(2^L)$. The problem remains, however, that $O(2^L)$ is a prohibitively large space for optimization. We therefore use a simple greedy algorithm for searching $S$. In the greedy algorithm, we start by initializing $S$ with all entries set to true. At each step of the greedy algorithm, we propose a neighboring pattern $S'$ by applying a bit-flip mutation, where each bit of $S$ is flipped with a probability of $\epsilon$. The proposal is accepted only if the acquisition value $a^*(\boldsymbol{u}^0, S')$ is higher than the current value $a^*(\boldsymbol{u}^0, S)$. This process of proposal and acceptance/rejection is repeated $T$ times. We use $T = 100$ and $\epsilon = 0.1$ throughout our experiments. A more concise summary of the batch acquisition algorithm is given in Appendix D.2. The algorithmic complexity of batch acquisition is discussed in Section 5.7.

## 4 RELATED WORK

**AL for PDEs.** The works by Pestourie et al. (2020); Pickering et al. (2022); Gajjar et al. (2022) apply active learning to problems involving PDEs, but their tasks are limited to predicting QoI, such as the maximum value of an evolved state. Li et al. (2024); Wu et al. (2023b) apply their AL methods to single-state prediction. Bajracharya et al. (2024) explores the use of active learning in tasks of predicting steady states of PDEs, which can be seen as predicting single states at $t \to \infty$. Finally, Musekamp et al. (2024) experiments with active learning in predicting PDE trajectories with autoregressive models.

**Active selection of time points.** While our work is the first to propose time step selection in active learning (AL) for PDEs, the concept of selecting time points has been explored in other contexts. For example, in physics-informed neural networks (PINNs), active selection of collocation points for training has been widely studied (Arthurs & King, 2021; Gao & Wang, 2023; Mao & Meng, 2023; Wu et al., 2023a; Turinici, 2024). "Labels" for PINNs, or the residual loss, can be calculated directly at any time point using closed form equations. There are also methods in Bayesian experimental design (BED) that choose observation times that maximize information gain about parameters of interest (Singh et al., 2005; Cook et al., 2008). In those works, a trajectory is already "there", but the cost is attributed to the act of observing a time point. In contrast, in our setting, we cannot directly acquire a time point, because there is a cost in the simulation of the trajectory.

**Multi-fidelity AL.** Closely related to our work is multi-fidelity active learning Li et al. (2022b); Wu et al. (2023b); Hernandez-Garcia et al. (2023); Li et al. (2024), where outputs are acquired at varying fidelity levels for each input, with associated costs inherent to each fidelity. In our context, the task of actively selecting a sampling pattern for a given initial condition can be seen as a fidelity selection problem, where acquiring all time steps corresponds to the highest fidelity but also incurs the highest cost.

## 5 EXPERIMENTS

### 5.1 BASELINE AL METHODS

To compare with our method, we experiment with AL for full trajectory sampling introduced in Musekamp et al. (2024). **Random** sampling from the pool set is the simplest method. **QbC** (Seung et al., 1992) is a simple active learning algorithm that selects points according to maximum disagreement among members of a committee. **LCMD** (Holzmüller et al., 2023) is an AL algorithm that uses a feature map. We concatenate the last hidden layer activations of committee members at all time steps of a trajectory, and sketch the concatenated features to a dimension of 512 using a random projection. Kirsch et al. (2023) proposes **SBAL**, which randomly samples data points $x$ with a probability distribution proportional to its temperature-scaled acquisition value $p(x) \propto a(x)^m$. We use the acquisition function of QbC with temperature $m = 1$. We leave out Core-Set (Sener & Savarese, 2017) because it generally underperforms compared to the above methods, according to both Holzmüller et al. (2023) and Musekamp et al. (2024).

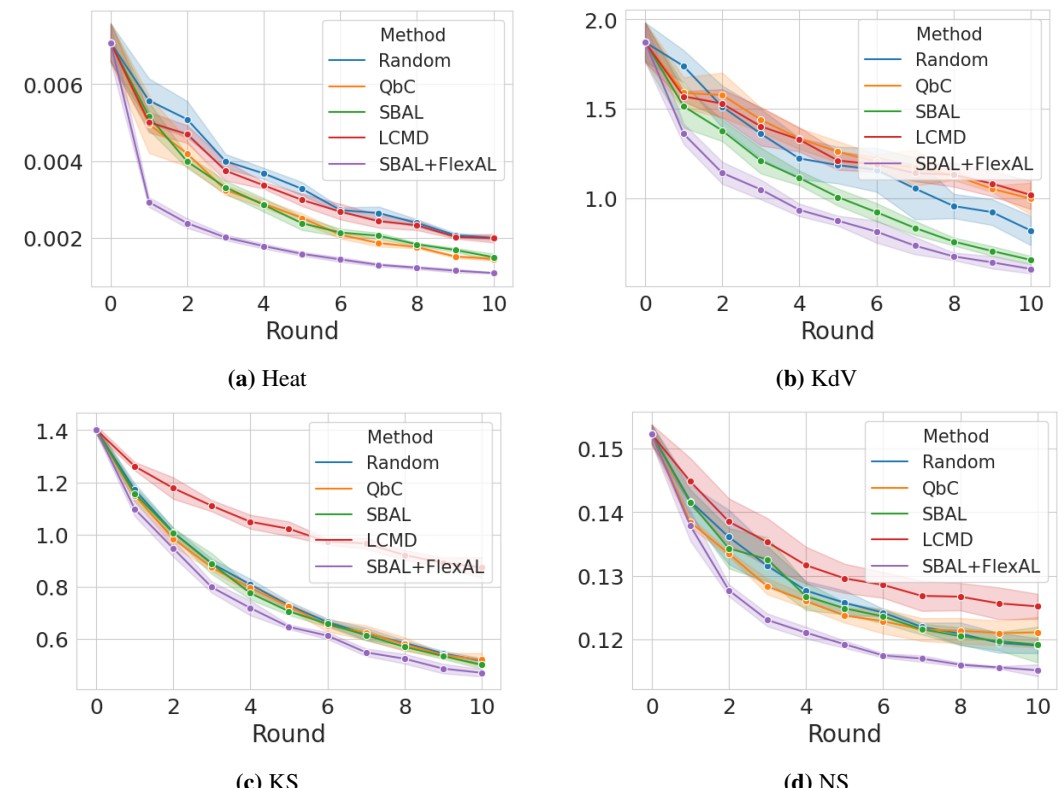

**Figure 3:** RMSE of AL strategies, measured across 10 rounds of acquisition. Each round incurs constant cost of data acquisition, namely the budget $B$.

## 5.2 TARGET PDEs

We evaluate our method on a range of PDEs. The first is the **Heat** equation in one spatial dimension. Next, we test the nonlinear Korteweg–De Vries (**KdV**) equation, which is known for exhibiting solitary wave pulses with weak interactions (Zabusky & Kruskal, 1965). We then apply our method to the Kuramoto–Sivashinsky (**KS**) equation, another nonlinear PDE in one dimension, notable for its chaotic dynamics. Lastly, we consider the vorticity form of the incompressible Navier-Stokes equation (**NS**) in two spatial dimensions. All equations are solved with periodic boundary conditions. Additional details are in Appendix A.1.

## 5.3 SURROGATE MODELS

We use a Fourier Neural Operator (FNO, Li et al., 2020) to model the evolution operator $G$. In particular, we train it to predict the differences between states in adjacent time steps, following Musekamp et al. (2024). All models have four hidden layers. We use 16, 256, 128, and 32 modes for Heat, KdV, KS, and NS equations, respectively. We also normalize the data according to the initial dataset's mean and standard deviation over all temporal and spatial dimensions. We use teacher-forcing to train the FNOs, meaning that it's simply trained on ground truth input-output pairs from the solver $G$ without backpropagating through two or more time steps. All models were trained with Adam (Kingma, 2014) for 100 epochs, using a learning rate of $10^{-3}$, a batch size of 32, and a cosine annealing scheduler (Loshchilov & Hutter, 2016).

## 5.4 RESULTS

We compare between the four baselines introduced in Section 5.1, and our method combined with SBAL (SBAL+FLEXAL). The pool set has 10,000 initial conditions, and we always start with an initial dataset of 32 fully sampled trajectories. The initial conditions in the test set are sampled from

**Table 1:** Log RMSE of baseline methods and SBAL+FLEXAL, averaged across 10 rounds of acquisition

|  | Random | QbC | LCMD | SBAL | SBAL+FLEXAL |
|---|---|---|---|---|---|
| **Heat** | $-5.688_{\pm 0.021}$ | $-5.924_{\pm 0.025}$ | $-5.741_{\pm 0.024}$ | $-5.901_{\pm 0.017}$ | $\mathbf{-6.304}_{\pm 0.015}$ |
| **KdV** | $0.191_{\pm 0.058}$ | $0.266_{\pm 0.027}$ | $0.256_{\pm 0.030}$ | $0.030_{\pm 0.029}$ | $\mathbf{-0.088}_{\pm 0.040}$ |
| **KS** | $-0.258_{\pm 0.003}$ | $-0.268_{\pm 0.003}$ | $0.046_{\pm 0.013}$ | $-0.275_{\pm 0.014}$ | $\mathbf{-0.349}_{\pm 0.003}$ |
| **NS** | $-2.050_{\pm 0.011}$ | $-2.057_{\pm 0.009}$ | $-2.018_{\pm 0.017}$ | $-2.052_{\pm 0.009}$ | $\mathbf{-2.092}_{\pm 0.003}$ |

**Table 2:** Log RMSE of FLEXAL averaged across 10 rounds, and their improvement over base methods. $\Delta$ refers to the improvements from baselines. Negative $\Delta$ indicates better performance of FLEXAL.

|  | Random | | QbC | | LCMD | |
|---|---|---|---|---|---|---|
|  | +FLEXAL | $\Delta$ | +FLEXAL | $\Delta$ | +FLEXAL | $\Delta$ |
| **Heat** | $-6.193 \pm 0.021$ | $-0.505 \pm 0.030$ | $-6.195 \pm 0.015$ | $-0.271 \pm 0.029$ | $-6.131 \pm 0.024$ | $-0.390 \pm 0.034$ |
| **KdV** | $-0.067 \pm 0.054$ | $-0.258 \pm 0.079$ | $0.134 \pm 0.035$ | $-0.132 \pm 0.044$ | $0.286 \pm 0.034$ | $0.030 \pm 0.045$ |
| **KS** | $-0.335 \pm 0.012$ | $-0.077 \pm 0.012$ | $-0.331 \pm 0.013$ | $-0.063 \pm 0.013$ | $-0.138 \pm 0.016$ | $-0.184 \pm 0.021$ |
| **NS** | $-2.080 \pm 0.003$ | $-0.030 \pm 0.011$ | $-2.079 \pm 0.008$ | $-0.022 \pm 0.012$ | $-2.050 \pm 0.007$ | $-0.032 \pm 0.018$ |

the same distribution as those in the pool set. An ensemble size of $M = 2$ is used, as it has been shown to be sufficient for good AL performance (Pickering et al., 2022; Musekamp et al., 2024). We perform 10 rounds of acquisition, and the budget of each round is set to $B = 8 \times L$ where $L$ is the length of a trajectory. This means that full trajectory algorithms sample 8 trajectories per round. We report their RMSE, defined in Appendix A.2. Reports of other metrics are provided in Appendix B. Fig. 3 shows plots of the committee's mean RMSE across the 10 rounds of acquisition, and Table 1 summarizes the results with mean logarithmic RMSEs, where a mean is taken over all 10 rounds. We can observe from the plots that SBAL+FLEXAL outperforms other AL baselines in a robust manner. Most notably, it improves the surrogate models on both the KS and NS equations, where no other baseline improves significantly over random sampling. On NS, SBAL+FLEXAL achieves an RMSE below 0.12 at the fifth round, which is only achieved by the best baseline at the tenth round. FLEXAL has effectively halved the cost of acquisition required to obtain this accuracy. All experiments were conducted on 8 NVIDIA GeForce RTX 2080 Ti GPUs, and the results are averages from 5 seed values.

## 5.5 OTHER BASE METHODS

We also report the mean log RMSE of FLEXAL when combined with the three other base methods, in Table 2. We find that the performance always improves over the base method with the addition of FLEXAL, except for the case of LCMD on KdV where the difference is negligible. Fig. 4 shows three plots of RMSE on the NS equation, where each contains a base method and its combination with FLEXAL. We find that the discrepancy between the two are noticeably larger for base methods that did not perform robustly when used alone. For instance, QbC tends to perform worse than Random in the later rounds, which is also where the discrepancy between QbC and QbC+FLEXAL becomes more noticeable. Also, LCMD performs the worst when used alone, and also creates the largest improvement when FLEXAL is added. We can infer that adding FLEXAL has the effect of swinging back to some loss curve, and that this effect is stronger for base methods that deviate more from it. We do note, however, that the loss curves of FLEXAL are distinct for different base methods.

## 5.6 RANDOM BERNOULLI SAMPLING OF TIME STEPS

We plot in Fig. 5 the distribution of time steps that our method chooses. The plot clearly shows the general tendency of FLEXAL to acquire the early time steps, with an occasional selection of the later time steps. The distributions still show clear differences between tasks, such as in their average number of time steps per trajectory or the frequency of later time steps. These suggests that FLEXAL is choosing time steps in an adaptive manner that's different for each task at hand.

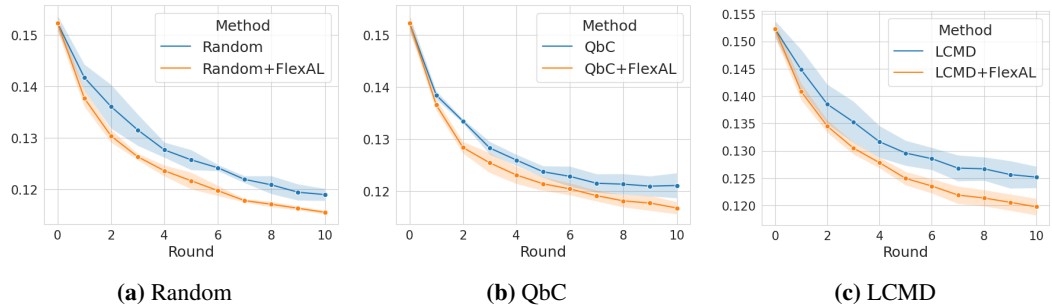

**(a)** Random          **(b)** QbC          **(c)** LCMD

**Figure 4:** RMSE of base methods with and without FLEXAL on NS, measured across 10 rounds of acquisition

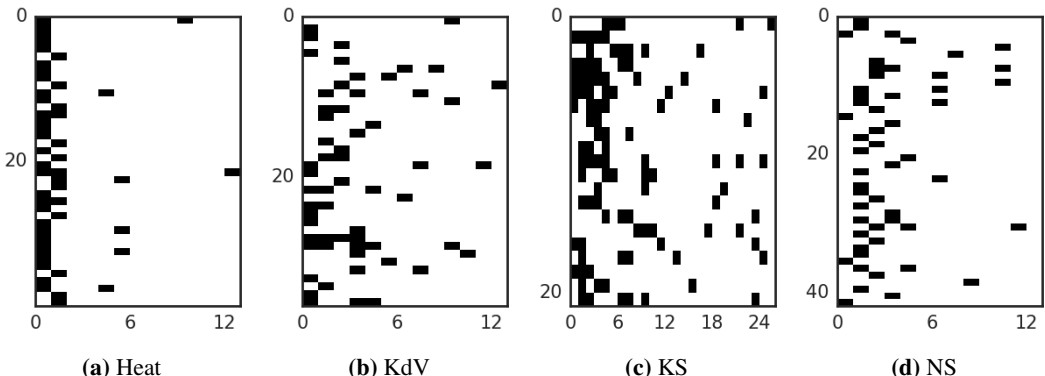

**(a)** Heat      **(b)** KdV      **(c)** KS      **(d)** NS

**Figure 5:** Timesteps chosen by SBAL+FLEXAL. Each row corresponds to an acquired trajectory, where the black cells indicate the selected time steps. Half of all trajectories acquired in the first rounds of active learning are shown.

We then ask ourselves: what if we perform random selection, for instance with a probability $p$, for every time step? We call this method Bernoulli sampling, or $\mathrm{Ber}(p)$, where each entry of $S$ is true with probability $p$. Table 3 summarizes the performance of $\mathrm{Ber}(p)$ for $p = 1/16$, 1/8, 1/4, and 1/2. Results show that FLEXAL outperforms Bernoulli sampling, except for the case of KS where $\mathrm{Ber}(1/16)$ serves as a strong alternative. In general, Bernoulli sampling improves over the base method SBAL, but it can also severely underperform at certain values of $p$, such as for KdV. Still, for each PDE, there exists a value of $p$ at which Bernoulli sampling provides an advantage over the base method SBAL. These observations show altogether that sparse sampling of time steps itself has an inherent advantage over full-trajectory sampling, and that FLEXAL amplifies this gain by adaptively choosing not only the frequency of the time steps to acquire, but also their locations. We report the full results in Appendix C.3, along with a variant of Bernoulli sampling that enforces acquiring consecutive initial time steps.

## 5.7 ALGORITHMIC COMPLEXITY OF FLEXAL

The time complexity of computing our acquisition function for a single instance of $(\boldsymbol{u}^0, S)$ is $O(M^2L)$. Since we optimize the acquisition function with $T$ steps, and we can acquire at most $B$ initial conditions, the time complexity of our batch acquisition algorithm is $O(M^2LBT)$ in the worst case. We can parallelize the optimization of multiple $S_j$'s to a certain extent using graphics processing unit (GPU), which can significantly alleviate the burden of $B$. We can further reduce the cost by at most a factor of $M$ with FLEXAL MF described in Appendix A.4. Yet another alternative is to decrease the number of greedy optimization steps $T$ from 100 to 10, which reduces the cost by a factor of 10. We call this variant FLEXAL 10. The wall-clock time of each baseline method and FLEXAL is summarized in Table 4. The performance of SBAL with FLEXAL and its two variants are summarized in Appendix C.4, as well as the wall-clock times on all equations. Note

**Table 3:** Log RMSE with Bernoulli sampling averaged across 10 rounds of acquisition

|  | SBAL | +FLEXAL | +Ber$(1/16)$ | +Ber$(1/8)$ | +Ber$(1/4)$ | +Ber$(1/2)$ |
|---|---|---|---|---|---|---|
| **Heat** | $-5.901_{\pm 0.017}$ | $\mathbf{-6.304}_{\pm 0.015}$ | $-6.093_{\pm 0.018}$ | $-6.071_{\pm 0.020}$ | $-6.057_{\pm 0.026}$ | $-6.010_{\pm 0.035}$ |
| **KdV** | $0.030_{\pm 0.029}$ | $\mathbf{-0.088}_{\pm 0.040}$ | $0.053_{\pm 0.014}$ | $0.049_{\pm 0.014}$ | $0.018_{\pm 0.024}$ | $-0.064_{\pm 0.031}$ |
| **KS** | $-0.275_{\pm 0.014}$ | $-0.349_{\pm 0.003}$ | $\mathbf{-0.365}_{\pm 0.008}$ | $-0.359_{\pm 0.006}$ | $-0.346_{\pm 0.008}$ | $-0.324_{\pm 0.007}$ |
| **NS** | $-2.052_{\pm 0.009}$ | $\mathbf{-2.092}_{\pm 0.003}$ | $-2.088_{\pm 0.005}$ | $-2.081_{\pm 0.008}$ | $-2.079_{\pm 0.007}$ | $-2.075_{\pm 0.009}$ |

**Table 4:** Wall-clock time of each procedure during batch selection in NS. Measured with a single NVIDIA GeForce RTX 2080 Ti GPU. Note that these are not the costs of data acquisition, but the computational cost of batch selection algorithms.

|  | Random | QbC | LCMD | SBAL | +FLEXAL | +FLEXAL MF | +FLEXAL 10 |
|---|---|---|---|---|---|---|---|
| Time taken (seconds) | $0.1_{\pm 0.1}$ | $45.5_{\pm 0.2}$ | $72.2_{\pm 1.4}$ | $45.1_{\pm 0.2}$ | $92.2_{\pm 2.6}$ | $55.9_{\pm 0.4}$ | $10.5_{\pm 1.2}$ |

that FLEXAL 10 incurs only a fraction of computational cost over the baseline methods, while still achieving a significant boost in performance over its base method. After all, the increased computational cost of the selection process is negligible in practical settings because the cost of data acquisition usually far exceeds the cost of selection. In fact, without running the numerical solver in batch mode, obtaining data for a single round in the KdV experiment takes *around 20 minutes*, which is far greater than any of the costs incurred by the selection algorithms. Moreover, increasing the pool size increases the runtime of base methods, but doesn't incur any additional runtime on FLEXAL.

## 6 CONCLUSION

In this paper, we presented a novel framework for active learning in surrogate modeling of partial differential equation (PDE) trajectories, significantly reducing the cost of data acquisition while maintaining or improving model accuracy. By selectively querying only a subset of time steps in a PDE trajectory, our method FLEXAL enables the acquisition of informative data at a fraction of the cost of acquiring entire trajectories. We introduced a new acquisition function that estimates the utility of a set of time steps based on variance reduction, effectively guiding the selection process in an adaptive manner. Through extensive experiments on benchmark PDEs, including the Heat equation, Korteweg–De Vries equation, Kuramoto–Sivashinsky equation, and incompressible Navier-Stokes equation, we demonstrated that our approach consistently outperforms existing AL methods, providing a more cost-efficient and accurate solution for PDE surrogate modeling.

Our results show that FLEXAL can significantly enhance surrogate modeling in PDEs, particularly in scenarios where the numerical solver is computationally expensive. We further showed that the success of FLEXAL is driven by its ability to prioritize both diverse and informative time steps. Moving forward, this framework could be extended to more complex systems and integrated with other machine learning techniques, providing broader applicability in scientific and engineering simulations. Future work may also explore alternative acquisition functions and applications to simulations outside the domain of PDEs.

REPRODUCIBILITY STATEMENT.

We present detailed description of our algorithm in Section 3.3. Details regarding the algorithm's hyperparameters, model architecture, training, active learning procedure, and data generation are provided in Section 5 and Appendix A.

ETHICS STATEMENT.

We propose a new method that improves the cost efficiency of acquiring data for building a surrogate model of PDE trajectories. Although our approach doesn't have a direct positive or negative impact in ethical or societal aspects, it accelerates the process of building a surrogate model for an arbitrary PDE. This could be used for good, such as medical simulations, environmental modeling, and optimizing engineering designs, potentially leading to advancements in healthcare, sustainability, and technological innovation. However, like many technologies, this method could also be misused in domains where rapid simulations could have harmful consequences, such as the development of hazardous materials. Therefore, researchers and practitioners should apply these methods with consideration of their broader societal implications, aiming to ensure that the benefits of the technology are used responsibly and ethically.

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

## A EXPERIMENTAL DETAILS

### A.1 DETAILS ON PDEs

In this section, we describe the PDEs used in our experiments. Each of these equations plays a critical role in modeling physical phenomena and showcases diverse behaviors, from diffusion and soliton dynamics to chaotic systems and fluid flow. Examples of PDE trajectories are shown in Fig. 6.

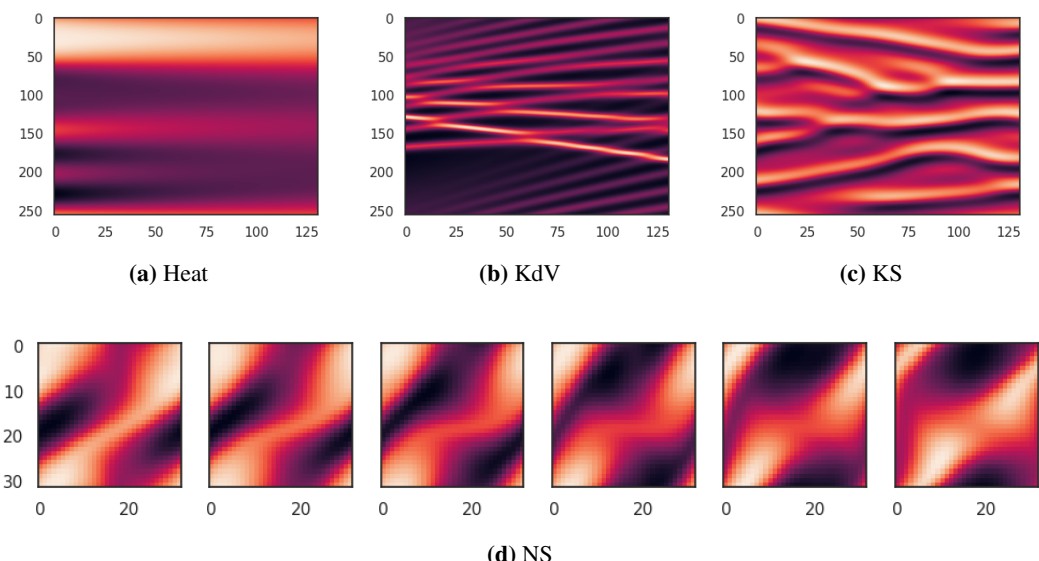

**(a)** Heat      **(b)** KdV      **(c)** KS

**(d)** NS

**Figure 6:** Example trajectories of PDEs. (a), (b), (c): Horizontal and vertical axes represent the temporal and spatial domain. (d): Two-dimensional states at six time points are shown.

**Heat Equation**    The one-dimensional (1D) Heat equation is given by:

$$\partial_t u = \partial_{xx} u, \tag{7}$$

where $u = u(x, t)$ represents the temperature distribution as a function of space $x$ and time $t$. This equation describes the process of heat conduction and diffusion in a medium. The simplicity of the Heat equation makes it a fundamental model for understanding diffusion-like processes across various fields in science and engineering, such as thermal conduction, population dynamics, and chemical diffusion. For our experiments, we solve this equation using the pseudospectral method with Dormand–Prince solver as in Brandstetter et al. (2022a).

**Korteweg–De Vries (KdV) Equation**    The second equation we study is the Korteweg–De Vries (KdV) equation, given by:

$$\partial_t u + u \partial_x u + \partial_{xxx} u = 0, \tag{8}$$

where $u = u(x, t)$ represents a wave profile evolving over space and time. This nonlinear PDE describes the evolution of shallow water waves, and its most famous characteristic is the presence of solitons—solitary, stable wave packets that maintain their shape over long distances and weak interactions with other waves (Zabusky & Kruskal, 1965). Solitons have important applications in fluid dynamics, plasma physics, and optical fiber communications. The KdV equation's nonlinearity and third-order spatial derivative ($\partial_{xxx}$) allow it to capture complex wave behavior. The equation is also known for conserving key quantities like energy. We solve this equation using the pseudospectral method with Dormand–Prince solver as in Brandstetter et al. (2022a).

**Kuramoto–Sivashinsky (KS) Equation**    The Kuramoto–Sivashinsky (KS) equation is a fourth-order nonlinear PDE, written as:

$$\partial_t u + \partial_{xx} u + \partial_{xxxx} u + u \partial_x u = 0, \tag{9}$$

**Table 5:** Domain lengths and discretizations for trajectory learning.

| PDE | Domain Length $(T, X)$ | Resolution $(L, N_x)$ |
|-----|------------------------|------------------------|
| Heat | (13.0, 6.28) | (13, 256) |
| KdV | (52.0, 128.0) | (13, 256) |
| KS | (13.0, 1.0) | (26, 256) |
| NS | (13.0, 1.0, 1.0) | (13, 32, 32) |

where $u = u(x, t)$ is the evolving field in space and time. The KS equation is known for its chaotic behavior and is used to model phenomena such as flame front propagation, plasma instabilities, and thin film dynamics. Its chaotic nature arises from the interplay between destabilizing nonlinear terms and stabilizing higher-order diffusion terms. The equation is particularly challenging to solve due to its sensitivity to initial conditions and long-term unpredictability. To handle this complexity, we use the Exponential Time Differencing (ETD) fourth-order Runge-Kutta method, as introduced by Kassam & Trefethen (2005). This numerical method is well-suited for stiff PDEs like the KS equation.

**Navier-Stokes (NS) Equation** The final equation we consider is the vorticity form of the incompressible Navier-Stokes (NS) equation, which governs the motion of viscous fluid flows. In two spatial dimensions, the vorticity formulation is given by:

$$\partial_t u + \boldsymbol{v} \cdot \nabla u = \nu \nabla^2 u + f, \quad \nabla \cdot \boldsymbol{v} = 0, \tag{10}$$

where $u(x_1, x_2, t)$ is the vorticity, $\boldsymbol{v}$ is the velocity field, $\nu$ is the kinematic viscosity, and $f(x_1, x_2)$ is an external forcing term. The Navier-Stokes equations describe the behavior of incompressible fluid flow, playing a central role in understanding turbulence, weather patterns, and aerodynamics. The external forcing term $f(x_1, x_2)$ is set to

$$f(x) = 0.1 \left( \sin(2\pi(x_1 + x_2)) + \cos(2\pi(x_1 + x_2)) \right), \tag{11}$$

which injects energy into the system, driving complex fluid dynamics. In our experiments, we adapt the Crank–Nicolson method implemented by Li et al. (2020).

**Initial conditions** As per Brandstetter et al. (2022a), states are first sampled from a simple distribution and then evolved for a certain time to obtain the initial conditions. The evolved initial conditions are more realistic than the sampled states, in that they are more likely to be observed under a system governed by the respective PDEs. This procedure hence approximates applications where the initial conditions of interest are realistic states either from observed data (Jumper et al., 2021; Kalnay, 2003; Chassignet et al., 2007; Taylor et al., 2012) or carefully crafted synthetic data (Jarrin et al., 2006; Kusner et al., 2017). For 1D equations, the states are sampled from truncated Fourier series with random coefficients (Brandstetter et al., 2022a), and for the 2D NS equation, states are sampled from a Gaussian random field as described in Li et al. (2020). The lengths and discretizations of trajectories are summarized in Table 5.

## A.2 ERROR METRICS

The test set always consists of 1,000 trajectories, on which several error metrics are defined. The **RMSE** is defined on a trajectory $\boldsymbol{u}$ as

$$\sqrt{\frac{1}{LN_x} \sum_{i=1}^{L} \sum_{j=1}^{N_x} \|\boldsymbol{u}^i(\mathbf{x}_j) - \hat{\boldsymbol{u}}^i(\boldsymbol{x}_j)\|_2^2}. \tag{12}$$

Similarly, the **NRMSE** is defined as

$$\sqrt{\frac{\sum_{i,j} \|\boldsymbol{u}^i(\mathbf{x}_j) - \hat{\boldsymbol{u}}^i(\boldsymbol{x}_j)\|_2^2}{\sum_{i,j} \|\boldsymbol{u}^i(\boldsymbol{x}_j)\|_2^2}} \tag{13}$$

**Table 6:** Acquired datasize in KdV

| Round | 0 | 1 | 2 | 3 | 4 | 5 | 6 | 7 | 8 | 9 | 10 |
|---|---|---|---|---|---|---|---|---|---|---|---|
| SBAL | 416 | 520 | 624 | 728 | 832 | 936 | 1040 | 1144 | 1248 | 1352 | 1456 |
| SBAL+FLEXAL | 416 | 507 | 611 | 715 | 819 | 923 | 1027 | 1131 | 1235 | 1339 | 1443 |

and the **MAE** as

$$\frac{1}{LN_x} \sum_{i=1}^{L} \sum_{j=1}^{N_x} |\boldsymbol{u}^i(\mathbf{x}_j) - \hat{\boldsymbol{u}}^i(\boldsymbol{x}_j)|. \tag{14}$$

The metrics are averaged across all trajectories in the test set. We also report their logarithmic values averaged across all AL rounds, following Holzmüller et al. (2023). Note that we do not use a committee's mean prediction for computing the metrics, but instead compute the metrics for each model and report their average.

### A.3 SIMULATION INSTABILITY

It was observed that using FLEXAL on the KdV equation, the simulation crashes on a small subset of synthetic inputs. Analysis reveals that these synthetic inputs have unusually large norms and particularly appear in later parts of trajectories due to accumulated error. We do not attempt to fix this problem explicitly due to the risk of over-complicating our method, and simply refrain from adding these time steps to the training dataset. This means that FLEXAL actually acquires a smaller number of time steps than the budget $B$ per round of acquisition, which could be problematic when a large subset of inputs do crash. However, we find that this is not the case, and the number of such inputs is small enough that FLEXAL can outperform other baselines. We report the comparison of datasize across rounds in Table 6, for a single experiment. We can see that 13 time steps were left out in the first round due to instability, and no instability occurred in the rounds after.

Since queries that crash incur a cost, they should be avoided as much as possible. Previous works in Bayesian optimization (Gelbart et al., 2014; Hernández-Lobato et al., 2015) propose methods to learn these unknown constraints. Alternatively, one could simply test out large, random inputs. In fact, we find that the maximum absolute value of an input being above 10 is a robust criterion for predicting that the solver will crash. Either way, we could simply filter out time steps that fall outside of these constraints during runtime of the solver, and use the freed up budget on acquiring other trajectories. Another possible approach is to impose physical constraints on the surrogate model (Goswami et al., 2022) that reduces the risk of outputting abnormal synthetic inputs. For instance, the KdV equation is energy-conserving, and when this prior knowledge is encoded into the surrogate model, the synthetic inputs would never be abnormally large like we experienced with our naive surrogate models.

### A.4 FLEXAL MF

We can also define a simpler acquisition function in the spirit of mean-field approximation. We take the mean model $\hat{G} = \frac{1}{M} \sum_m \hat{G}_m$, and define the variance reduction $R(\hat{G}, b, S)$ between $\hat{G}$ and a model $\hat{G}_b$ in the same way as before. We then average the variance reduction between the mean model and all models in the committee:

$$a_{\text{MF}}(\boldsymbol{u}^0, S) = \frac{1}{M} \sum_{b \in [M]} R(\hat{G}, b, S), \tag{15}$$

which reduces the computational cost by a factor of $M$ in the best case. We call this modified version FLEXAL MF.

## B FULL REPORT OF RESULTS ON MAIN EXPERIMENT

We provide a full report of all results from the main experiment. Table 7, Table 8, Table 9, Table 10 show the full results on Heat, KdV, KS, and NS equations, respectively. Fig. 7 shows the plots of RMSE quantiles on all PDEs. Fig. 8 shows the plots of NRMSE on all PDEs.

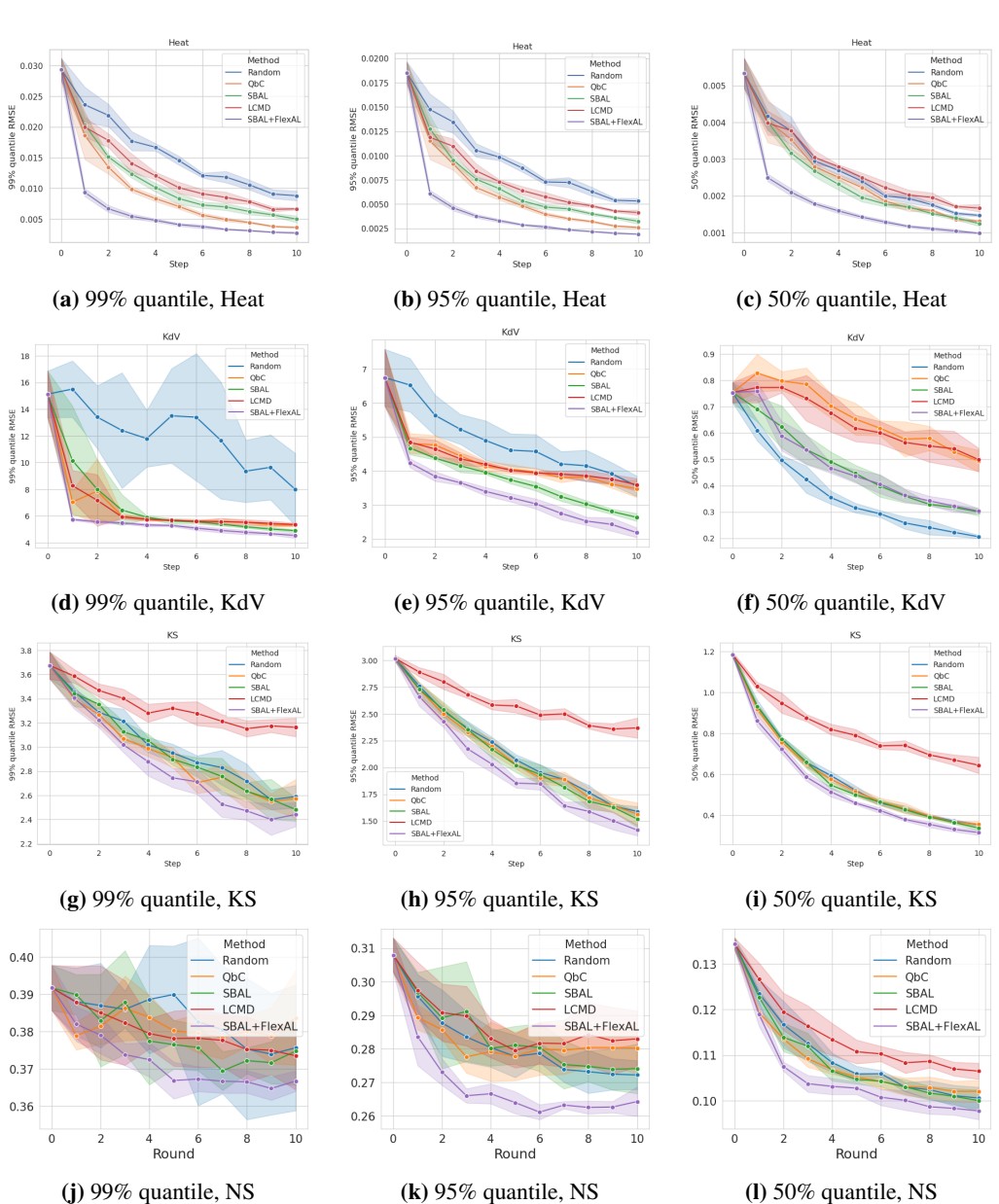

**Figure 7:** Mean logarithmic values of RMSE quantiles

**Table 7:** Mean log metrics for Heat Equation

|  | RMSE | NRMSE | MAE | 99% | 95% | 50% |
|---|---|---|---|---|---|---|
| Random | $-5.688_{\pm0.021}$ | $-6.486_{\pm0.018}$ | $-7.486_{\pm0.021}$ | $-4.211_{\pm0.029}$ | $-4.712_{\pm0.022}$ | $-5.992_{\pm0.020}$ |
| SBAL | $-5.901_{\pm0.017}$ | $-6.644_{\pm0.015}$ | $-7.699_{\pm0.018}$ | $-4.624_{\pm0.049}$ | $-5.073_{\pm0.035}$ | $-6.111_{\pm0.008}$ |
| LCMD | $-5.741_{\pm0.024}$ | $-6.494_{\pm0.023}$ | $-7.541_{\pm0.024}$ | $-4.466_{\pm0.027}$ | $-4.945_{\pm0.026}$ | $-5.940_{\pm0.023}$ |
| QbC | $-5.924_{\pm0.025}$ | $-6.637_{\pm0.025}$ | $-7.724_{\pm0.024}$ | $-4.848_{\pm0.027}$ | $-5.222_{\pm0.024}$ | $-6.068_{\pm0.024}$ |
| SBAL+FLEXAL | $-6.304_{\pm0.015}$ | $-7.014_{\pm0.015}$ | $-8.114_{\pm0.015}$ | $-5.284_{\pm0.020}$ | $-5.653_{\pm0.012}$ | $-6.433_{\pm0.014}$ |
| Random+FLEXAL | $-6.193_{\pm0.021}$ | $-6.953_{\pm0.017}$ | $-7.997_{\pm0.021}$ | $-4.862_{\pm0.037}$ | $-5.348_{\pm0.032}$ | $-6.426_{\pm0.016}$ |
| QbC+FLEXAL | $-6.195_{\pm0.015}$ | $-6.880_{\pm0.018}$ | $-8.008_{\pm0.015}$ | $-5.401_{\pm0.010}$ | $-5.654_{\pm0.013}$ | $-6.273_{\pm0.017}$ |
| LCMD+FLEXAL | $-6.132_{\pm0.024}$ | $-6.843_{\pm0.023}$ | $-7.944_{\pm0.024}$ | $-5.147_{\pm0.033}$ | $-5.512_{\pm0.027}$ | $-6.247_{\pm0.023}$ |

**Table 8:** Mean log metrics for KdV Equation

|  | RMSE | NRMSE | MAE | 99% | 95% | 50% |
|---|---|---|---|---|---|---|
| Random | $0.191_{\pm0.058}$ | $-1.193_{\pm0.050}$ | $-2.034_{\pm0.045}$ | $2.449_{\pm0.047}$ | $1.395_{\pm0.049}$ | $-1.196_{\pm0.043}$ |
| SBAL | $0.030_{\pm0.029}$ | $-1.282_{\pm0.030}$ | $-2.139_{\pm0.027}$ | $1.875_{\pm0.039}$ | $1.267_{\pm0.028}$ | $-1.267_{\pm0.027}$ |
| QbC | $0.266_{\pm0.027}$ | $-1.019_{\pm0.029}$ | $-1.879_{\pm0.029}$ | $1.859_{\pm0.037}$ | $1.251_{\pm0.029}$ | $-1.019_{\pm0.031}$ |
| LCMD | $0.256_{\pm0.030}$ | $-1.033_{\pm0.036}$ | $-1.879_{\pm0.033}$ | $1.868_{\pm0.034}$ | $1.322_{\pm0.034}$ | $-1.100_{\pm0.038}$ |
| SBAL+FLEXAL | $-0.088_{\pm0.040}$ | $-1.378_{\pm0.040}$ | $-2.239_{\pm0.043}$ | $1.731_{\pm0.040}$ | $1.280_{\pm0.043}$ | $-1.378_{\pm0.040}$ |
| Random+FLEXAL | $-0.067_{\pm0.054}$ | $-1.425_{\pm0.047}$ | $-2.228_{\pm0.036}$ | $1.885_{\pm0.033}$ | $1.296_{\pm0.038}$ | $-1.424_{\pm0.044}$ |
| QbC+FLEXAL | $0.134_{\pm0.035}$ | $-1.130_{\pm0.037}$ | $-2.004_{\pm0.035}$ | $1.721_{\pm0.031}$ | $1.120_{\pm0.037}$ | $-1.286_{\pm0.035}$ |
| LCMD+FLEXAL | $0.286_{\pm0.034}$ | $-0.978_{\pm0.034}$ | $-1.824_{\pm0.039}$ | $1.799_{\pm0.036}$ | $1.128_{\pm0.034}$ | $-1.129_{\pm0.032}$ |

Following Holzmüller et al. (2023), we also report the 99%, 95%, and 50% quantiles of RMSE. This is useful for analyzing the behavior of AL strategies. AL methods tend to improve performance on points with extreme errors, thus improving performance significantly in the top quantiles, while not so much in the middle quantiles. This is why AL methods perform differently depending on the nature of problem. For instance, problems with more irregularities tend to benefit signficantly more from AL methods, since the top quantile errors contribute significantly to the average error in those problems.

As expected, the baseline methods improve performance over random sampling in the 99% quantile, but not so much in the 95% and 50% quantiles. Suprisingly, FLEXAL robustly outperforms the baselines in all error quantiles, which is rarely the case for existing AL methods. We can therefore infer that FLEXAL isn't simply sacrificing the surrogate model's performance in some trajectories to improve its performance in others. FLEXAL both sees a more diverse set of trajectories, and samples the most informative time steps in each trajectory, effectively accounting for how it can improve performance in both the high and middle quantiles of error.

## C  ADDITIONAL EXPERIMENTS

### C.1  DIVERSITY OF SPARSELY SELECTED TIME STEPS

We provide a simple analysis to show that time steps sampled in a sparse manner are more diverse than time steps from entire trajectories. Out of 128 trajectories, we first randomly chose 10 trajectories, which contains $L \times 10$ states. Then, out of all $L \times 128$ states, we randomly chose $L \times 10$ states. The first choice represents full trajectory sampling, and the latter represents spare time steps sampling. We probe an FNO surrogate model trained on all the 128 trajectories at its hidden layer, and observe the hidden layer activation at each of the $L \times 128$ states. The result is shown in Fig. 9, where black points represent states from the fully sampled trajectories and red points represent sparsely selected states. The latter states are visibly more diverse, which partially explains how sampling time steps in a sparse manner from trajectories can benefit a surrogate model.

**Table 9:** Mean log metrics for KS Equation

| | RMSE | NRMSE | MAE | 99% | 95% | 50% |
|---|---|---|---|---|---|---|
| Random | $-0.258_{\pm0.004}$ | $-1.683_{\pm0.004}$ | $-2.165_{\pm0.004}$ | $1.097_{\pm0.003}$ | $0.752_{\pm0.005}$ | $-0.575_{\pm0.004}$ |
| SBAL | $-0.275_{\pm0.014}$ | $-1.700_{\pm0.014}$ | $-2.184_{\pm0.014}$ | $1.086_{\pm0.017}$ | $0.732_{\pm0.023}$ | $-0.594_{\pm0.012}$ |
| QbC | $-0.268_{\pm0.004}$ | $-1.693_{\pm0.004}$ | $-2.178_{\pm0.004}$ | $1.077_{\pm0.008}$ | $0.739_{\pm0.013}$ | $-0.582_{\pm0.006}$ |
| SBAL+FLEXAL | $-0.349_{\pm0.003}$ | $-1.774_{\pm0.003}$ | $-2.265_{\pm0.003}$ | $1.042_{\pm0.011}$ | $0.672_{\pm0.012}$ | $-0.674_{\pm0.008}$ |
| Random+FLEXAL | $-0.335_{\pm0.014}$ | $-1.759_{\pm0.014}$ | $-2.248_{\pm0.014}$ | $1.060_{\pm0.015}$ | $0.691_{\pm0.007}$ | $-0.662_{\pm0.015}$ |
| QbC+FLEXAL | $-0.331_{\pm0.014}$ | $-1.756_{\pm0.014}$ | $-2.246_{\pm0.014}$ | $1.050_{\pm0.013}$ | $0.681_{\pm0.020}$ | $-0.650_{\pm0.013}$ |
| LCMD | $0.046_{\pm0.015}$ | $-1.378_{\pm0.015}$ | $-1.829_{\pm0.015}$ | $1.204_{\pm0.009}$ | $0.954_{\pm0.011}$ | $-0.203_{\pm0.016}$ |
| LCMD+FLEXAL | $-0.138_{\pm0.017}$ | $-1.561_{\pm0.016}$ | $-2.033_{\pm0.017}$ | $1.139_{\pm0.006}$ | $0.841_{\pm0.014}$ | $-0.431_{\pm0.016}$ |

**Table 10:** Mean log metrics for NS Equation

| | RMSE | NRMSE | MAE | 99% | 95% | 50% |
|---|---|---|---|---|---|---|
| SBAL | $-2.052_{\pm0.009}$ | $-4.253_{\pm0.009}$ | $-4.592_{\pm0.008}$ | $-0.970_{\pm0.011}$ | $-1.260_{\pm0.019}$ | $-2.217_{\pm0.010}$ |
| Random | $-2.050_{\pm0.011}$ | $-4.249_{\pm0.011}$ | $-4.590_{\pm0.010}$ | $-0.959_{\pm0.029}$ | $-1.266_{\pm0.011}$ | $-2.208_{\pm0.016}$ |
| QbC | $-2.057_{\pm0.009}$ | $-4.258_{\pm0.009}$ | $-4.597_{\pm0.009}$ | $-0.962_{\pm0.006}$ | $-1.261_{\pm0.019}$ | $-2.217_{\pm0.011}$ |
| LCMD | $-2.018_{\pm0.017}$ | $-4.219_{\pm0.017}$ | $-4.560_{\pm0.016}$ | $-0.967_{\pm0.017}$ | $-1.248_{\pm0.022}$ | $-2.168_{\pm0.019}$ |
| SBAL+FLEXAL | $-2.092_{\pm0.003}$ | $-4.293_{\pm0.003}$ | $-4.632_{\pm0.003}$ | $-0.988_{\pm0.005}$ | $-1.309_{\pm0.004}$ | $-2.249_{\pm0.011}$ |
| Random+FLEXAL | $-2.080_{\pm0.003}$ | $-4.280_{\pm0.003}$ | $-4.621_{\pm0.003}$ | $-0.980_{\pm0.008}$ | $-1.303_{\pm0.005}$ | $-2.235_{\pm0.007}$ |
| QbC+FLEXAL | $-2.079_{\pm0.008}$ | $-4.280_{\pm0.008}$ | $-4.619_{\pm0.008}$ | $-0.979_{\pm0.003}$ | $-1.293_{\pm0.017}$ | $-2.238_{\pm0.005}$ |
| LCMD+FLEXAL | $-2.051_{\pm0.007}$ | $-4.253_{\pm0.007}$ | $-4.593_{\pm0.007}$ | $-0.984_{\pm0.011}$ | $-1.258_{\pm0.009}$ | $-2.211_{\pm0.013}$ |

## C.2 REGULARIZATION FOR TRAJECTORY LEARNING

Brandstetter et al. (2022b) identifies a potential problem with training an autoregressive surrogate model with teacher-forcing. The model experiences a distribution shift during inference, because errors accumulate during rollout unlike during training. They propose a simple fix, called the push-forward trick, which supervises the model $\hat{G}$ not with pairs of $\boldsymbol{u}^{i-1}$ and $\boldsymbol{u}^i$, but with pairs of $\hat{G}[\boldsymbol{u}^{i-2}]$ and $\boldsymbol{u}^i$, where $\hat{G}$ is constantly changing throughout training. An even simpler fix that they experiment with is augmenting the inputs with a Gaussian noise.

One might hypothesize that the advantage of FLEXAL comes from its regularizing effect, since the synthetic inputs in the training set are outputs from the surrogate model $\hat{G}$. We therefore apply the pushforward trick and Gaussian noise augmentation on the best performing baseline method, SBAL. The results in Table 11 shows that the effect of such regularization methods is minimal compared to the effect of FLEXAL. This shows that the advantage of FLEXAL lies not just in its regularizing effect.

## C.3 RANDOM BERNOULLI SAMPLING OF TIME STEPS

We provide the whole list of results with Bernoulli sampling described in Section 5.6. Also, we can enforce consecutive initial time steps sampling by bringing all the true entries in $S$ to the beginning. We call this method Initial Bernoulli sampling, or Initial Ber($p$). We report the results with SBAL in Table 12 and Table 13. Initial Bernoulli sampling always performs the worst, possibly because they rarely see the time steps at the end.

## C.4 EFFICIENT VARIANTS OF FLEXAL

We provide results with two efficient variants of FLEXAL, namely FLEXAL MF and FLEXAL 10. The results are summarized in Table 14. We provide the wall-clock times of selection algorithms in all equations, in Table 15.

We haven't done extensive experiments with different values of $T$ and $\epsilon$. Increasing $T$ improves performance until it plateaus. Increasing $\epsilon$ values higher than a certain point deteriorates the performance slightly. On the other hand, decreasing $\epsilon$ too much also deteriorates the performance, but can be recovered with a higher value of $T$, leading to higher computational cost.

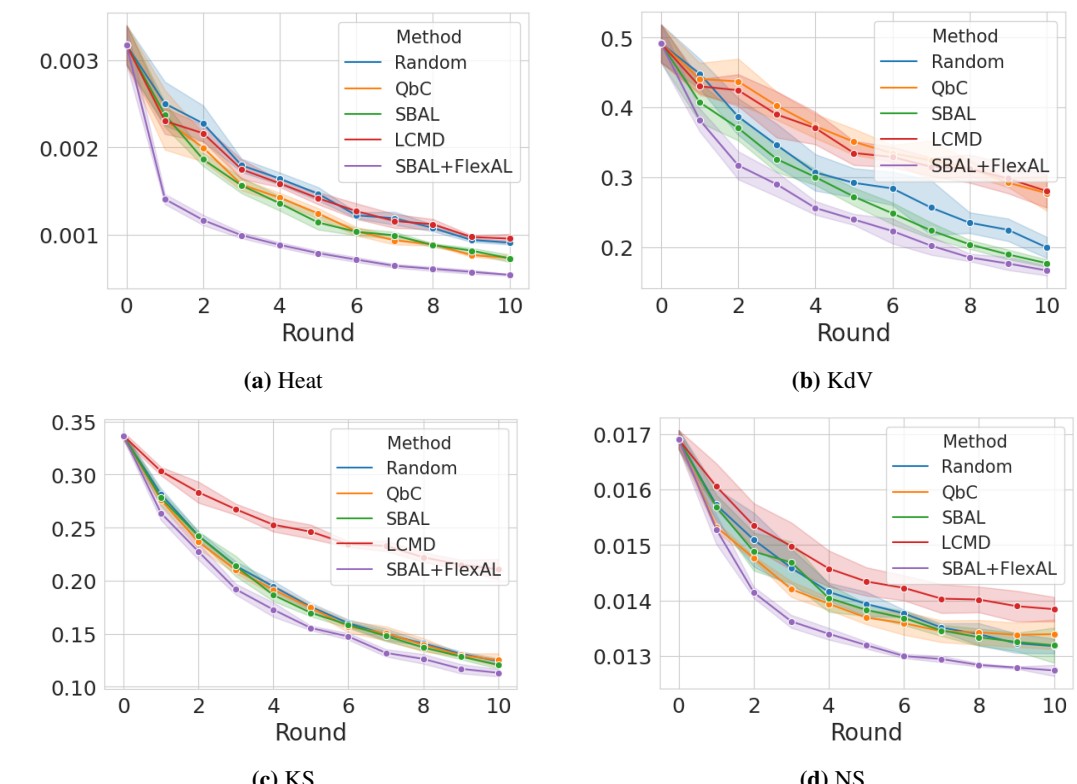

**(a)** Heat

**(b)** KdV

**(c)** KS

**(d)** NS

**Figure 8:** NRMSE of AL strategies, measured across 10 rounds of acquisition. Each round incurs constant cost of data acquisition, namely the budget $B$. These are simply scaled versions of the RMSE plots.

## D FURTHER EXPLANATION OF ACQUISITION WITH FLEXAL

### D.1 MOTIVATION BEHIND THE ACQUISITION FUNCTION

Here we detail the motivation behind our acquisition function defined in Section 3.2. First, one can imagine several alternative acquisition functions.

The most straightforward alternative is to simply use the sum of the variances at time points for which $b_i = $ true. The variances are larger for the later time steps since they accumulate, and in our preliminary experiments, we found that this is catastrophic as undersampling the earlier time steps leads to the sampled trajectory being very out-of-distribution, and hence the trained surrogate model underperforming on the test distribution.

It quickly became clear to us that we need some kind of measure of "how much total uncertainty will be reduced by sampling these time steps", instead of "how uncertain is our model on these time steps?" This would help select sampling patterns that reduce the out-of-distribution-ness introduced by $\hat{G}$. One way to approximate this is to use mutual information, as used by Li et al. (2022b). In other words, we would rollout $M$ trajectories with $M$ surrogate models, and compute the mutual information between time steps for which $b_i = $ true and all time steps. However, in preliminary experiments, we found that this method underperforms, which we hypothesize is because relying simply on the covariance matrix of the committee between time steps is not a good enough method for computing the posterior uncertainty.

We identified two "pathways" through which sampling a time step reduces uncertainty in the remaining time steps. First, there is the "indirect" pathway: sampling a time step will reduce the model's uncertainty on similar inputs, hence reducing uncertainty on the remaining time steps. This is what is approximated by mutual information. Then, there is the "direct" pathway: sampling a time step

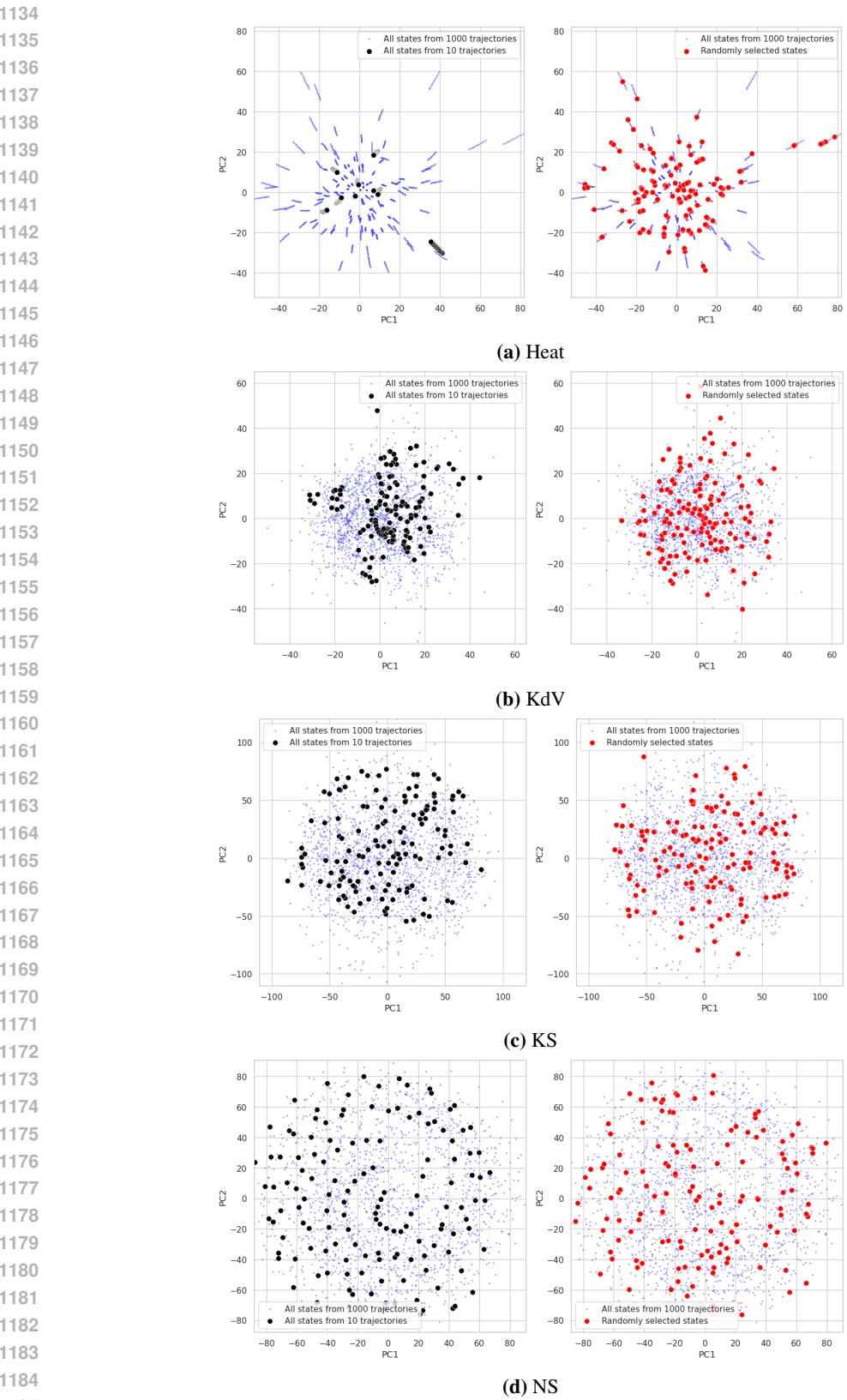

**Figure 9:** PCA of FNO hidden layer's activation pattern for both entire trajectories (black) and sparsely sampled time steps (red)

**Table 11:** Effect of regularization

|  | SBAL | +FLEXAL | +Pushforward | +Gaussian |
|---|---|---|---|---|
| **Heat** | | | | |
| RMSE | $-5.901_{\pm0.017}$ | $\mathbf{-6.304}_{\pm0.015}$ | $-3.086_{\pm1.584}$ | $-5.844_{\pm0.011}$ |
| NRMSE | $-6.644_{\pm0.015}$ | $\mathbf{-7.014}_{\pm0.015}$ | $-3.755_{\pm1.627}$ | $-6.558_{\pm0.013}$ |
| MAE | $-7.699_{\pm0.018}$ | $\mathbf{-8.114}_{\pm0.015}$ | $-4.884_{\pm1.583}$ | $-7.630_{\pm0.012}$ |
| **KdV** | | | | |
| RMSE | $0.030_{\pm0.029}$ | $\mathbf{-0.088}_{\pm0.040}$ | $0.924_{\pm0.613}$ | $0.017_{\pm0.042}$ |
| NRMSE | $-1.282_{\pm0.030}$ | $\mathbf{-1.378}_{\pm0.040}$ | $-0.245_{\pm0.689}$ | $-1.292_{\pm0.042}$ |
| MAE | $-2.139_{\pm0.027}$ | $\mathbf{-2.239}_{\pm0.043}$ | $-1.077_{\pm0.690}$ | $-2.162_{\pm0.041}$ |
| **KS** | | | | |
| RMSE | $-0.275_{\pm0.014}$ | $\mathbf{-0.349}_{\pm0.003}$ | $1.148_{\pm0.795}$ | $-0.259_{\pm0.012}$ |
| NRMSE | $-1.700_{\pm0.014}$ | $\mathbf{-1.774}_{\pm0.003}$ | $-0.283_{\pm0.792}$ | $-1.684_{\pm0.012}$ |
| MAE | $-2.184_{\pm0.014}$ | $\mathbf{-2.265}_{\pm0.003}$ | $-0.473_{\pm0.956}$ | $-2.167_{\pm0.012}$ |
| **NS** | | | | |
| RMSE | $-2.052_{\pm0.009}$ | $\mathbf{-2.092}_{\pm0.003}$ | $-0.060_{\pm1.118}$ | $-2.067_{\pm0.012}$ |
| NRMSE | $-4.253_{\pm0.009}$ | $\mathbf{-4.293}_{\pm0.003}$ | $-2.258_{\pm1.119}$ | $-4.267_{\pm0.012}$ |
| MAE | $-4.592_{\pm0.008}$ | $\mathbf{-4.632}_{\pm0.003}$ | $-2.619_{\pm1.107}$ | $-4.606_{\pm0.012}$ |

**Table 12:** Bernoulli sampling

|  | SBAL | +FLEXAL | +Ber(1/16) | +Ber(1/8) | +Ber(1/4) | +Ber(1/2) |
|---|---|---|---|---|---|---|
| **Heat** | | | | | | |
| RMSE | $-5.901_{\pm0.017}$ | $\mathbf{-6.304}_{\pm0.015}$ | $-6.093_{\pm0.018}$ | $-6.071_{\pm0.020}$ | $-6.057_{\pm0.026}$ | $-6.010_{\pm0.035}$ |
| NRMSE | $-6.644_{\pm0.015}$ | $\mathbf{-7.014}_{\pm0.015}$ | $-6.823_{\pm0.019}$ | $-6.801_{\pm0.020}$ | $-6.791_{\pm0.027}$ | $-6.748_{\pm0.033}$ |
| MAE | $-7.699_{\pm0.018}$ | $\mathbf{-8.114}_{\pm0.015}$ | $-7.893_{\pm0.019}$ | $-7.872_{\pm0.019}$ | $-7.858_{\pm0.027}$ | $-7.810_{\pm0.034}$ |
| **KdV** | | | | | | |
| RMSE | $0.030_{\pm0.029}$ | $\mathbf{-0.088}_{\pm0.040}$ | $0.053_{\pm0.014}$ | $0.049_{\pm0.014}$ | $0.018_{\pm0.024}$ | $-0.064_{\pm0.031}$ |
| NRMSE | $-1.282_{\pm0.030}$ | $\mathbf{-1.378}_{\pm0.040}$ | $-1.254_{\pm0.017}$ | $-1.257_{\pm0.014}$ | $-1.288_{\pm0.020}$ | $-1.370_{\pm0.033}$ |
| MAE | $-2.139_{\pm0.027}$ | $\mathbf{-2.239}_{\pm0.043}$ | $-2.082_{\pm0.016}$ | $-2.083_{\pm0.018}$ | $-2.120_{\pm0.025}$ | $-2.207_{\pm0.034}$ |
| **KS** | | | | | | |
| RMSE | $-0.275_{\pm0.014}$ | $-0.349_{\pm0.003}$ | $\mathbf{-0.365}_{\pm0.008}$ | $-0.359_{\pm0.006}$ | $-0.346_{\pm0.008}$ | $-0.324_{\pm0.007}$ |
| NRMSE | $-1.700_{\pm0.014}$ | $-1.774_{\pm0.003}$ | $\mathbf{-1.790}_{\pm0.008}$ | $-1.784_{\pm0.006}$ | $-1.771_{\pm0.008}$ | $-1.749_{\pm0.007}$ |
| MAE | $-2.184_{\pm0.014}$ | $-2.265_{\pm0.003}$ | $\mathbf{-2.282}_{\pm0.007}$ | $-2.276_{\pm0.006}$ | $-2.262_{\pm0.009}$ | $-2.237_{\pm0.009}$ |
| **NS** | | | | | | |
| RMSE | $-2.052_{\pm0.009}$ | $\mathbf{-2.092}_{\pm0.003}$ | $-2.088_{\pm0.005}$ | $-2.081_{\pm0.008}$ | $-2.079_{\pm0.007}$ | $-2.075_{\pm0.009}$ |
| NRMSE | $-4.253_{\pm0.009}$ | $\mathbf{-4.293}_{\pm0.003}$ | $-4.288_{\pm0.005}$ | $-4.282_{\pm0.008}$ | $-4.279_{\pm0.007}$ | $-4.276_{\pm0.009}$ |
| MAE | $-4.592_{\pm0.008}$ | $\mathbf{-4.632}_{\pm0.003}$ | $-4.626_{\pm0.004}$ | $-4.620_{\pm0.008}$ | $-4.617_{\pm0.007}$ | $-4.614_{\pm0.009}$ |

$i$ gives out the $i+1$ th state, which starts a chain reaction of reducing model uncertainty on all successive states. Note that these two pathways are not distinct from a strictly theoretical view, but are rather two ways of approximating uncertainty reduction.

The direct pathway motivated our acquisition function based on variance reduction. In variance reduction, we calculate the posterior uncertainty by rolling out the trajectories with $N$ surrogate models, but collapse into one surrogate model at time steps for which $b_i =$ true. This effectively computes the reduced uncertainty due to the effect of the direct pathway. With experiments, we confirmed that this acquisition function behaves just like we wanted: it is slightly biased towards sampling the earlier time steps, and it chooses an appropriate frequency of time steps to sample that leads to good performance.

**Table 13:** Initial Bernoulli sampling

|  | Initial Ber$(1/16)$ | Initial Ber$(1/8)$ | Initial Ber$(1/4)$ | Initial Ber$(1/2)$ |
|---|---|---|---|---|
| **Heat** | | | | |
| RMSE | $-6.278_{\pm 0.016}$ | $-6.254_{\pm 0.015}$ | $-6.182_{\pm 0.019}$ | $-6.080_{\pm 0.017}$ |
| NRMSE | $-6.989_{\pm 0.014}$ | $-6.966_{\pm 0.014}$ | $-6.902_{\pm 0.018}$ | $-6.811_{\pm 0.017}$ |
| MAE | $-8.088_{\pm 0.016}$ | $-8.062_{\pm 0.014}$ | $-7.987_{\pm 0.019}$ | $-7.881_{\pm 0.017}$ |
| **KdV** | | | | |
| RMSE | $0.032_{\pm 0.016}$ | $-0.015_{\pm 0.014}$ | $-0.001_{\pm 0.018}$ | $0.011_{\pm 0.014}$ |
| NRMSE | $-1.278_{\pm 0.014}$ | $-1.321_{\pm 0.017}$ | $-1.303_{\pm 0.018}$ | $-1.294_{\pm 0.014}$ |
| MAE | $-2.150_{\pm 0.016}$ | $-2.197_{\pm 0.014}$ | $-2.181_{\pm 0.018}$ | $-2.168_{\pm 0.014}$ |
| **KS** | | | | |
| RMSE | $-0.302_{\pm 0.009}$ | $-0.293_{\pm 0.008}$ | $-0.287_{\pm 0.005}$ | $-0.283_{\pm 0.009}$ |
| NRMSE | $-1.728_{\pm 0.009}$ | $-1.719_{\pm 0.008}$ | $-1.713_{\pm 0.005}$ | $-1.708_{\pm 0.009}$ |
| MAE | $-2.216_{\pm 0.008}$ | $-2.206_{\pm 0.008}$ | $-2.199_{\pm 0.007}$ | $-2.194_{\pm 0.010}$ |
| **NS** | | | | |
| RMSE | $-2.045_{\pm 0.016}$ | $-2.044_{\pm 0.014}$ | $-2.051_{\pm 0.019}$ | $-2.058_{\pm 0.014}$ |
| NRMSE | $-4.246_{\pm 0.014}$ | $-4.244_{\pm 0.014}$ | $-4.251_{\pm 0.019}$ | $-4.258_{\pm 0.014}$ |
| MAE | $-4.596_{\pm 0.016}$ | $-4.594_{\pm 0.014}$ | $-4.598_{\pm 0.019}$ | $-4.602_{\pm 0.014}$ |

**Table 14:** Log RMSE of more efficient FLEXAL variants averaged across 10 rounds.

|  | SBAL | +FLEXAL | +FLEXAL MF | +FLEXAL 10 |
|---|---|---|---|---|
| **Heat** | $-5.901_{\pm 0.017}$ | $\mathbf{-6.304}_{\pm 0.015}$ | $-6.303_{\pm 0.009}$ | $-6.058_{\pm 0.020}$ |
| **KdV** | $0.030_{\pm 0.029}$ | $\mathbf{-0.088}_{\pm 0.040}$ | $-0.065_{\pm 0.034}$ | $-0.118_{\pm 0.024}$ |
| **KS** | $-0.275_{\pm 0.014}$ | $\mathbf{-0.349}_{\pm 0.003}$ | $-0.326_{\pm 0.004}$ | $-0.316_{\pm 0.009}$ |
| **NS** | $-2.052_{\pm 0.009}$ | $-2.092_{\pm 0.003}$ | $\mathbf{-2.093}_{\pm 0.004}$ | $-2.078_{\pm 0.010}$ |

## D.2 BATCH ACQUISITION ALGORITHM

Algorithm 2 summarizes the batch selection algorithm of FLEXAL. Starting with an empty batch $\mathcal{B}$, the algorithm repeatedly selects initial conditions and their sampling patterns until reaching the budget limit. It first uses the base active learning method $\mathcal{A}$ to choose an initial condition $\boldsymbol{u}^0$. Then, it optimizes which time steps to sample through a greedy procedure: starting with a pattern $S$ that samples all time steps (all true values), it performs $T$ iterations of random mutations. In each iteration, it generates a candidate pattern $S'$ by randomly flipping entries in $S$ with probability $\epsilon$ (using a binary mask $C$ where each entry is drawn from a Bernoulli distribution and the XOR operation $\oplus$). If this new pattern achieves a better value according to the cost-weighted acquisition function $a^*$, it becomes the current pattern. To ensure the budget isn't exceeded, if adding the current pattern would go over budget, the algorithm truncates it by keeping only enough true values to exactly meet the budget. The pair of initial condition and its optimized sampling pattern $(u^0, S)$ is then added to the batch $\mathcal{B}$.

| Equation | QbC | +FlexAL | +FlexAL 10 |
|---|---|---|---|
| Heat | 10.3 | 43.0 | 4.3 |
| KdV | 10.6 | 40.1 | 4.5 |
| KS | 18.1 | 78.4 | 8.6 |
| NS | 45.5 | 92.2 | 10.5 |

**Table 15:** Wall-clock times of selection algorithms for all equations

---

**Algorithm 2** Batch Acquisition Algorithm

---

**Require:** Budget $B$, base active learning algorithm $\mathcal{A}$, probability $\epsilon$, number of iterations $T$ for greedy optimization, pool $P$ of initial conditions, cost function $\text{cost}(\cdot)$ for batches.
**Ensure:** A batch $\mathcal{B}$ of initial conditions and sampling patterns.
1: $\mathcal{B} \leftarrow \varnothing$
2: **while** $\text{cost}(\mathcal{B}) < B$ **do**
3:     Acquire an initial condition $\boldsymbol{u}^0$ with $\mathcal{A}$.
4:     Initialize $S \leftarrow (\text{true}, \dots, \text{true})$.
5:     **for** $i = 1$ to $T$ **do**
6:         $C = (C_1, \dots, C_L)$ where $C_1, \dots, C_L \overset{\text{i.i.d.}}{\sim} \text{Ber}(\varepsilon)$.
7:         $S' = S \oplus C$
8:         **if** $a^*(\boldsymbol{u}^0, S') \geq a^*(\boldsymbol{u}^0, S)$ **then**
9:             $S \leftarrow S'$.
10:         **end if**
11:     **end for**
12:     **if** $\|S\| + \text{cost}(\mathcal{B}) > B$ **then**
13:         Keep only the first $(B - \text{cost}(\mathcal{B}))$ trues from $S$ and flip the remaining trues.
14:     **end if**
15:     $\mathcal{B} \leftarrow \mathcal{B} \cup \{(\boldsymbol{u}^0, S)\}$.
16: **end while**

---

