# OpenReview forum: "Flexible Active Learning of PDE Trajectories"
_ICLR.cc/2025/Conference — Submitted to ICLR 2025_

### Official Review · Reviewer_Y8Kg · 2024-10-22

**Soundness:** 3
**Presentation:** 3
**Contribution:** 2
**Rating:** 5
**Confidence:** 4

**Summary:**

The paper proposes a method of active learning for PDEs. When training a neural operator for PDE, we often generate data using solvers. Solvers can be expensive, and now the question is how we can develop an active learning method to efficiently sample from a solver, in particular for temporal PDEs.

This paper proposes to train an ensemble of models and use their disagreement as a sign to collect samples.

**Strengths:**

The paper tackles an important problem in practice.

**Weaknesses:**

There is a set of weaknesses in this work that the authors are encouraged to address.

1- The novelty. The method is very similar to prior works, with similar equitation function, ensemble approach, and philosophy.

2- One might be able to say that not much about PDE-ness is present in this approach. Similar approaches are also in the game engine, video simulation, and 3D scene settings. Nothing about PDEs is exploited here, nothing I can see it is about PDE-ness, so the approach is not dedicatedly designed for PDEs. So what is particular about this method which is related to PDE-ness of the problem?

3- Baseline active learning for simulation, video, and 3D scenes would be relevant.

4- The datasets are quite simple for this paper.

If the authors tried this approach on a challenging task, then the first 3 points, as well as the next point, could be ignored.

5- The method is quite ad-hoc and heuristic. What if all the models wrongly agree on many inputs and time steps? How can one guarantee this method doesn't collapse or capture the right "variance" and "uncertainty"? Just relying that this method, hopefully, does not collapse, is not sufficient. Again, this point could be ignored if we actually could tackle a challenging problem.

**Questions:**

minor, in Eq3, we often after expected value not the empirical loss.

---

> ### Comment · Area_Chair_7hnb · 2024-11-14
> **Further details**
>
> Dear Reviewer Y8Kg,
>
> could you provide further details on your review, in order to provide more concrete feedback to the authors and also to put your recommendation in perspective?
>
> Namely:
>
> - could you add references to claim that the method lacks novelty?
> - what do you mean with PDE-ness? How that plays a role in your criticism as it seems that this is method to train surrogate models.
> - could you provide an explicit recommendation for the baseline you suggest? (for simulation/image/3s scenes).
> - could you provide an example of a challenging problem?
>
> This type of actionable feedback would be great for the authors, and also to put your recommendation in perspective.
>
> Thank you again for your time and expertise.
>
> Best,
>
> AC

---

> ### Author Response · Authors · 2024-11-15
>
> We would like to clarify our paper's main contribution.
>
> Our core contribution is a novel framework for acquiring PDE trajectory data that allows sampling specific time steps rather than requiring entire trajectories. The key aspects are:
>
> 1. Regarding "method is similar to prior works":
> - This is the first work to propose partial trajectory sampling for PDEs
> - Unlike prior works that must acquire complete trajectories, we can selectively sample time steps
> - Our acquisition function is specifically designed to handle this novel sampling setting
> - We demonstrate clear improvements over baseline methods, e.g., achieving same accuracy with half the data acquisition cost
> - We highly suggest reading through **Common Response 1. Simple explanation** for a simple explanation of our method
>
> 2. Regarding "nothing about PDE-ness":
> - Our method explicitly leverages the Markov property of PDEs where each state depends only on the previous state
> - The sampling pattern design accounts for PDE-specific temporal dependencies
> - We demonstrate effectiveness on challenging PDEs including Navier-Stokes
> - While the general framework could potentially extend beyond PDEs, solving PDEs efficiently is an important problem with real-world applications
>
> 3. Regarding "ad-hoc and heuristic":
> - Uncertainty-based approaches have solid theoretical foundations [1] [2]
> - We provide empirical validation across multiple PDEs
> - The improvements are consistent across different error metrics and quantiles
>
> We hope this helps clarify our contribution. We welcome any specific questions about these aspects of our work.
>
> References:
>
> [1] Yoav Freund, H Sebastian Seung, Eli Shamir, and Naftali Tishby. Selective sampling using the query by committee algorithm. *Machine Learning*, 28:133–168, 1997.\
> [2] Steve Hanneke. Theoretical foundations of active learning. Carnegie Mellon University, 2009.

---

> ### Author Response · Authors · 2024-11-24
>
> We thank you again for the insightful questions and feedbacks. We have posted a response to your concerns and a revised version of our manuscript. As we reach the end of our discussion period, your response would be of great help in further improving our work.

---

> > ### Comment · Reviewer_Y8Kg · 2024-11-26
> > **PDE-Ness**
> >
> > I would like to thank the authors for their response. I will take it into account for the final recommendation.

---

### Official Review · Reviewer_WEGs · 2024-10-29

**Soundness:** 3
**Presentation:** 3
**Contribution:** 3
**Rating:** 8
**Confidence:** 3

**Summary:**

Design-of-experiments for training surrogate models of time series. If simulated "experiments" are expensive, then it might be cheaper to train the surrogates sparsely. More specifically, a committee of surrogates is trained to estimate the uncertainty, and then value of new samples, which are then actually sampled and used to train the model.

**Strengths:**

The idea seems natural; In a sense a lot of machine learning is simply working out which data points need to be observed to train the correct model. In operator learning in particular we suspect much of the effort in training neural operators is wasted.

This paper is nearly simple, which is IMO great.

**Weaknesses:**

Given that sparse adaptive sampling from simulators is such a natural idea, the authors' solution "feels" surprisingly contrived. Sampling is strictly by masking over a fixed-timestep simulation pattern; to my mind it would be more satisfying to search for useful initial conditions from the traiing distribution, or to simply give up on rollouts when a given trajectory was no longer adding useful value, or to learn a predictor which could be conditioned on a rollout time, and we might imagine some scheme more elegant than this Bernoulli masking. However, this is not a blocker. Maybe this paper outlines the best progress than can be made at the moment? Maybe the best solution is not that elegant. If so, no problem.

A bigger problem for me is that I have a hard time understanding how the rollout accounts for serial dependencies in the training data. I have done my best to try to understand it, but I suspect that this might be a deficiency in the authors' explanation itself. I would be prepared to revise my recommendation if they could make this part clearer. See Questions below for a lengthier series of questions on that theme. Without clarification on that I cannot really assess the later part of the paper.

`FlexAL` is not IMO a great name; this is not particularly flexible compare to other active learning/DOE methods.

**Questions:**

I'm having a hard time understanding how the causal dependencies in a simulator are accounted for in this method. Can you lay it out for me? ("Explain it like I'm 5")
AFAICT each successive step in the simulator depends on the previous ones. Figure 6 seems to support this, as does text l283 "in our setting, we cannot directly acquire a time point, because there is a cost in the simulation of the trajectory.").

Reasoning this through and attempting to crosscheck with the text, I haven't been able to work it out.
So if I "mask out" timestep $t$ but I wish to train on timestep $t+1$,  don't I still need to simulate at timestep $t$? How do I save the cost of the $t$th step at full fidelity? I get that we can roll out a surrogate or a solver, and that we can choose which to use, but when we want the actual "ground truth" solver, how do we get that? Does our actual roll-out incorporate mixtures of models, e.g. in a given training run we might have some "mixed" rollout like $\hat{u}^2=G \circ \hat{G}  u^0$? (here $\circ$ is composition) OK, that seems fine, but at a later stage of my training I would get a different output from $\hat{G}$, so my training data would change and then I would need to recalculate $\hat{u}^2$, right? This should introduce an asymmetry between the acquisition functions early in the time series, and later, since the later ones are likely to be sample from some other distribution than the training distribution (e.g. they may have failed to conserve mass or momentum or whatever, when the surrogate roll-out was used).

---

> ### Author Response · Authors · 2024-11-15
>
> We appreciate your thoughtful questions and feedbacks. We encourage you to read through **Common Response 1. Simple explanation** for some clarifications. Below we also answer the specific questions you had about the method. If these still do not answer your questions, or if you have any other questions, please feel free to ask us.
>
> > AFAICT each successive step in the simulator depends on the previous ones. Figure 6 seems to support this, as does text l283 "in our setting, we cannot directly acquire a time point, because there is a cost in the simulation of the trajectory.").
>
> That is right. As described in Section 3.1, we roll out the trajectory with a mixture of the numerical solver and the surrogate, and each successive step depends on the previous step.
>
> > OK, that seems fine, but at a later stage of my training I would get a different output from $\hat{G}$
>
> That is right.
>
> > so my training data would change and then I would need to recalculate $\hat{u}^2$, right?
>
> No, we just keep using the training data obtained in the earlier round, that is, $(x,y) = \left ( \hat G u^0, (G \circ \hat G)u^0  \right ) $ obtained with $\hat G$ in the earlier round. We also think our paper could benefit from some clarification on this, and plan to do so in a revised version. Thank you for the question.
>
> We emphasize that time steps simulated with $\hat G$, e.g. $(x,y) = (u^0, \hat G u^0)$, are never added to our training data, but only those simulated with $G$ (Section 3.1).
>
> > This should introduce an asymmetry between the acquisition functions early in the time series, and later, since the later ones are likely to be sample from some other distribution than the training distribution (e.g. they may have failed to conserve mass or momentum or whatever, when the surrogate roll-out was used).
>
> We designed the acquisition function, described in Section 3.2, to address this problem. Imagine an alternative acquisition function that simply selects time points with large variance (this is not what ours is doing). The variance would be larger for later time points because their inputs will be more out-of-distribution. Perhaps this is what you were imagining when asking this question. This would indeed be catastrophic, as the active learner will be heavily biased towards sampling the later time points, only worsening the problem. Our acquisition function is different, however, and estimates the reduction in variance. The acquisition function thus penalizes a sampling pattern $S$ that is expected to yield an out-of-distribution trajectory. We hope this answers your question, but if not, we would be more than happy to elaborate.

---

> ### Author Response · Authors · 2024-11-15
>
> > to my mind it would be more satisfying to search for useful initial conditions from the traiing distribution, or to simply give up on rollouts when a given trajectory was no longer adding useful value, or to learn a predictor which could be conditioned on a rollout time, and we might imagine some scheme more elegant than this Bernoulli masking
>
> We appreciate these ideas. Note that we do actually search for useful initial conditions from the training distribution, as described in Section 3.3. This is done by a base AL method $\mathcal A$ (e.g. SBAL) on which we attach FlexAL. As to your second idea, we actually started our research with the same idea, but found a critical problem with these kinds of approaches: they rarely sample the later time steps of a trajectory. This renders the surrogate model’s performance extremely low on the later parts of trajectories. (Note that the distribution of the PDE states varies across time points, so data at earlier time steps are not guaranteed to help the model in later time steps.) Table 12 and 13 in the Appendix provide partial proof of this claim, as sampling initial time steps only (Initial Ber(p)) gave surrogate models with much worse performance, sometimes even worse than full-trajectory sampling. We hence realized that sampling time steps almost uniformly is crucial to success. To do so, we need to be able to skip some intermediate time steps, which we think is only possible with the surrogate model. Despite our initial concern with the resulting shift in the input distribution, we empirically found that its effect is much smaller than the overall benefit of sampling from more trajectories under the same budget. On your last idea of a predictor conditioned on rollout time, this is an existing method in the literature of PDE modeling, but is not standard as it tends to underperform compared to the approach of modeling individual time steps. Overall, we aimed at the simplest method possible that can deliver true robust gains in performance.
>
> > FlexAL is not IMO a great name; this is not particularly flexible compare to other active learning/DOE methods.
>
> We also see how it might not have been the best name, as it can’t be applied to active learning outside of PDE data. We appreciate the feedback, and will consider changing the name to a more suitable form.

---

> > ### Comment · Reviewer_WEGs · 2024-11-15
> >
> > To be clear my score does not depend upon the authors renaming the method; I merely mention the name problem because I am reviewing many papers that name something non-specific "Flexible" or "Generalized" etc; whether the paper is easy to find on google should probably be the arbiter of the suitability of the name, not my personal taste.

---

> ### Comment · Reviewer_WEGs · 2024-11-15
>
> I think the authors for addressing my questions. This explanation has substantially improved my understanding of the work, and my estimation of its value. I am prepared to revise my score upwards conditional upon the final explanation in the paper clarifying the explanation and thus my confusion (which I observe I share with other reviewers).  The explanation in the common response is a good start; I think clarifying it to the satisfaction of the reviewers might need a little more intuition-building, and clarification of language throughout to disambiguate the various kinds of "Steps" etc. I think the authros have demonstrated good progress already in the review process already

---

> > ### Comment · Reviewer_WEGs · 2024-11-21
> > **New PDF is much better thank you**
> >
> > Revising my score; I may revise further after I have had time to revisit.
> >
> > > l507. In fact, without running the numerical solver in batch mode, obtaining data for a single round in the KdV experiment takes 1,232 seconds,
> >
> > Minor quibble: 4 significant figures feels like a lot of precision for a timing. Consider "obtaining data for a single round in the KdV experiment takes more than 20 minutes"?

---

> ### Author Response · Authors · 2024-11-15
>
> Thank you for your prompt reply. We are glad that the explanations helped in clarifying the confusions. We will update you with a revised version of the paper as soon as possible. In the meantime, please feel free to ask any further questions or comments if you have any.

---

> ### Author Response · Authors · 2024-11-21
>
> Thank you for the positive review.
>
> We agree with the suggestion and will change it in the next revised version.

---

### Official Review · Reviewer_NgwX · 2024-10-30

**Soundness:** 3
**Presentation:** 3
**Contribution:** 2
**Rating:** 8
**Confidence:** 4

**Summary:**

For a partial differential equation (PDE) $\partial_t u = F(u, \partial_x u)$, fixed time-interval $\Delta t$, and a solution/evolution operator $G$, which satisfies $G u(t, \cdot) \approx u(t+\Delta t, \cdot)$, the context for the submission is to approximate $G$ with a surrogate through active learning.
The submission proposes an alternative to computing full trajectories of the PDE solution and appending those to the dataset. Namely, the suggestion is to choose a sampling pattern S ~ (True, False, True, True, ...), simulate the PDE using a mix of surrogate and numerical solver (depending on True/False at each step), append all indices corresponding to the numerical solver to the dataset (skip the surrogate steps), retrain, and repeat.
The sampling pattern is chosen by optimising an acquisition function based on variance reduction, respectively, a batch version of it.


Now, the driving question for assessing this method is how much this "sparse" sampling pattern improves over full-trajectory (or initial-step) active learning (e.g., Musekamp et al., 2024). The manuscript investigates an answer to this question on typical PDE benchmark problems.

**Strengths:**

The proposed algorithm is a clear generalisation of existing PDE active-learning methods: Instead of acquiring full trajectories or initial conditions, any combination of time points can now be used.
The manuscript is generally easy to follow (I point out specific questions below, but these are really minor). The experiments are comprehensive, and the paper is in good shape overall.

**Weaknesses:**

The proposed algorithm generalises existing schemes through corresponding choices of sampling patterns.
However, optimising the sampling pattern is, simply put, a lot of work (concretely, $O(2^L)$ if one doesn't use the greedy selection algorithm, where $L$ is the sequence's length; Section 3.3 discusses this thoroughly). And while this additional work seems to improve the quality of the reconstructions (Tables 1, 2, & 3; Figures 3 & 4), the proposed FlexAL takes significantly longer to run (Table 4).
Whether or not this increased runtime is problematic likely depends on the PDE.
However, the increased runtime is a weakness of the proposed method compared to existing techniques nonetheless.

The submission acknowledges this shortcoming and discusses a fix that cuts the number of training epochs for FlexAL. Still, this discussion leaves some questions to be answered. For example, the training epochs could also be reduced for existing methods with corresponding runtime gains, and it needs to be clarified how this affects the results (unless I've missed something; I checked Appendix 5.8 and couldn't find such a discussion).

Now, to convince me that this weakness isn't one, it would suffice to find an example where the additional computation for batch selection (compared to full-trajectory QbC, for instance) can be negligibly small. Alternatively, it would be interesting to see what happens to the reconstruction results in Table 1 or 2 if all columns do not receive the same number of iterations but are limited to roughly the same wall time.

However, I understand that these changes are likely outside the scope of a revision, and I am in favour of accepting this paper without them. That said, if I get convinced that the assessed weakness isn't one, my score would be slightly higher.

**Questions:**

I group my questions into more important and less important ones.
I don't expect a reply to the less important ones, but I would appreciate some clarification on the more important ones.
The answers do not affect my rating. However, I believe the manuscript would improve if the paper included them.

More important:

- Section 3.2: _Why_ choose the acquisition function based on variance reduction? Are there other candidates, and if so, why are they less suitable?
- Table 1: log-RMSEs of $\approx 0.1$ imply RMSEs of $\approx 1.0$, which seems to be large. Is it fair to assume that the solvers learn anything reasonable? For example, what happens if one plots the solutions and compares the surrogate's solution to the solver's? Suppose the reconstruction is good despite these large errors. Could the table be made clearer (in the sense of "success" meaning "RMSE far below 1") by choosing (for example) relative RMSE over absolute RMSE?
Table 4: It would be great to include the context of how much runtime a numerical solver needs to simulate Navier-Stokes. Whichever outcome (in terms of "who's faster") is fine, but the context would help assess the efficiency of the active-learning methods. If the solver is slow, the context would underline the statement in line 034 that states how costly numerical solvers can be.


Less important:

- Line 053, "we argue that querying all the states (...) is not cost-efficient": does this sentence perhaps require more nuances? While sparse subsets of trajectories decrease the complexity (on paper), the suggested procedure for finding them is sufficiently expensive that the proposed algorithm is more costly than full-batch versions (Section 3.3, Table 4).
- Line 095: Perhaps Brandstetter et al. (2022b) are not the best reference for spatiotemporal PDEs. Personally, I don't think this sentence needs any reference, but if one should be used, perhaps something like Evans' "Partial Differential Equations" book would be more appropriate.
- Line 100: This statement about existence would benefit from a reference.
- Line 112: What does "primary focus" mean here? It seems to be the _only_ focus. Have I missed something?
- Line 128: Why does this sentence introduce a distribution of initial conditions, but the rest of the manuscript (for example, Algorithm 1) operates on a pool of initial conditions? I understand that the pool is sampled from the condition, but it might be more reader-friendly to use a pool of conditions throughout.
- Line 262: How do the results depend on this choice of $T$ and $\epsilon$?
- Line 298: The abbreviation "QbC" is used early in the paper (for instance, in line 117 or 204). Maybe it would be good to introduce it before line 298.

---

> ### Author Response · Authors · 2024-11-15
>
> We appreciate your thoughtful questions and feedbacks.
>
> > The submission acknowledges this shortcoming and discusses a fix that cuts the number of training epochs for FlexAL.
>
> We would like to clarify that FlexAL 10 in Section 5.8 is actually not a cut in the number of training epochs, but the number of optimization steps $T$ in the greedy selection algorithm. Table 14 in the Appendix shows that the performance of FlexAL 10 is comparable to that of FlexAL, and always outperforms the best baseline method.
>
> > However, the increased runtime is a weakness of the proposed method compared to existing techniques nonetheless.
>
> > Now, to convince me that this weakness isn't one, it would suffice to find an example where the additional computation for batch selection (compared to full-trajectory QbC, for instance) can be negligibly small.
>
> We appreciate this feedback. In practical settings, the cost of data acquisition (running the numerical solver) is so high that the cost of batch selection is almost negligible. For example, if we do not use batch processing for the numerical solver, acquiring KdV trajectories for one round of AL takes 1,232 seconds, while the batch selection process of QbC, FlexAL, and FlexAL 10 respectively take 10.6, 40.1, and 4.5 seconds. (With batch processing or for other benchmark equations, our numerical solver is too cheap to run that they don't really help in proving our point.)
>
> To directly address your concern about the increased runtime in batch selection process, we provide an example where FlexAL adds minimal runtime compared to the base method. If the pool size is ten times larger (10,000 -> 100,000) than used in our experiment, the wall clock time of QbC and SBAL increases tenfold, while that of FlexAL remains constant because it is proportional to the budget, not the pool size. This larger pool size of 100,000 is used in Musekamp et al. [1], and is more reflective of real world settings where unlabelled data are abundant. For KdV, the runtime of QbC would be around $10.6 \times 10 = 106$ seconds, while for FlexAL and FlexAL 10 it would remain constant at 40.1 and 4.5 seconds. A 4.5/106 ~= 4% increase in computational cost due to FlexAL 10 is negligible. The same analysis holds for all equations, as summarized in Table 1.
>
> Table 1. Expected wall-clock time of batch selection, with pool size 100,000.
>
> | Equation | QbC | +FlexAL | +FlexAL 10 | Ratio (+FlexAL 10 / QbC) |
> |---|---|---|---|---|
> | Heat | 103 | 43.0 | 4.3 | 0.04 |
> | KdV | 106 | 40.1 | 4.5 | 0.04 |
> | KS | 181 | 78.4 | 8.6 | 0.05 |
> | NS | 455 | 92.2 | 10.5 | 0.02 |
>
> (We used the smaller pool size of 10,000 due to memory constraints. The fact that wall-clock time of QbC increases proportionally with pool size was validated empirically with measurements at pool size of 100, 1000, and 10,000. The same was done for FlexAL and FlexAL 10 to check that their wall-clock times remain constant.)
>
> We also think these are important points that should be addressed in the paper, and plan to revise the manuscript accordingly.

---

> ### Author Response · Authors · 2024-11-15
>
> > Section 3.2: Why choose the acquisition function based on variance reduction? Are there other candidates, and if so, why are they less suitable?
>
> One can imagine several alternative acquisition functions.
>
> The most straightforward alternative is to simply use the sum of the variances at time points for which $b_i = \mathsf{true}$. The variances are larger for the later time steps since they accumulate, and in our preliminary experiments, we found that this is catastrophic as undersampling the earlier time steps leads to the sampled trajectory being very out-of-distribution, and hence the trained surrogate model underperforming on the test distribution.
>
> It quickly became clear to us that we need some kind of measure of "how much total uncertainty will be reduced by sampling these time steps", instead of "how uncertain is our model on these time steps?" One way to approximate this is to use mutual information (or information gain), as used by Li et al. [1]. In other words, we rollout $N$ trajectories with $N$ surrogate models, and compute the mutual information between time steps for which $b_i=\mathsf{true}$ and all time steps. However, in preliminary experiments, we found that this method underperforms, which we hypothesize is because relying simply on the covariance matrix of the committee between time steps is not a good enough method for computing the posterior uncertainty.
>
> We identified two "pathways" through which sampling a time step reduces uncertainty in the remaining time steps. First, there is the "indirect" pathway: sampling a time step will reduce the model's uncertainty on similar inputs, hence reducing uncertainty on the remaining time steps. This is what is approximated by mutual information. Then, there is the "direct" pathway: sampling a time step $i$ gives out the $i+1$ th state, which starts a chain reaction of reducing model uncertainty on all successive states. Note that these two pathways are not distinct from a strictly theoretical view, but are rather two ways of approximating uncertainty reduction.
>
> The direct pathway motivated our acquisition function based on variance reduction. In variance reduction, we calculate the posterior uncertainty by rolling out the trajectories with $N$ surrogate models, but collapse into one surrogate model at time steps for which $b_i = \mathsf{true}$. This effectively computes the reduced uncertainty due to the effect of the direct pathway. With experiments, we confirmed that this acquisition function performs robustly, and behaves just like we wanted: it is slightly biased towards sampling the earlier time steps, and it chooses an appropriate frequency of time steps to sample that leads to good performance.
>
> We will make sure to include a simplified version of this in our revised manuscript. Thank you for the question, we also think this is an important missing part of the current manuscript.
>
> References:
>
> [1] Shibo Li, Zheng Wang, Robert M. Kirby, and Shandian Zhe. Deep multi-fidelity active learning of
> high-dimensional outputs. In *International Conference on Artificial Intelligence and Statistics*,
> AISTATS 2022, 28-30 March 2022, Virtual Event, 2022b.

---

> ### Author Response · Authors · 2024-11-15
>
> > Table 1: log-RMSEs of  imply RMSEs of , which seems to be large. Is it fair to assume that the solvers learn anything reasonable? For example, what happens if one plots the solutions and compares the surrogate's solution to the solver's? Suppose the reconstruction is good despite these large errors. Could the table be made clearer (in the sense of "success" meaning "RMSE far below 1") by choosing (for example) relative RMSE over absolute RMSE?
>
> The RMSEs are on different scales for each PDE due to the inherent scales of the solutions. I have attached an instance of KS equation where the surrogate model has instance-specific RMSE of 0.958.
>
> https://anonymous.4open.science/api/repo/iclr2025rebuttal-18AB/file/ks.png?v=5f58a930
>
> As you suggest, normalized (relative) RMSE is a better measure of the "success" of a model that is scale-independent. In fact, the average log NRMSE of Random on KS is -1.683, compared to an average log RMSE of −0.258. We have followed the convention of Musekamp et al. [2] in using RMSE instead of normalized RMSE, but we now think that normalized RMSE would have been more appropriate. We already have tables of normalized RMSE in Appendix B, but not the graphs of normalized RMSE against AL round. We will add these graphs. We will also provide instances of the surrogate model's predictions like the attached image.
>
> References:
>
> [2] Daniel Musekamp, Marimuthu Kalimuthu, David Holzmüller, Makoto Takamoto, and Mathias Niepert. Active learning for neural pde solvers. *arXiv preprint arXiv:2408.01536*, 2024.

---

> ### Author Response · Authors · 2024-11-15
>
> > Table 4: It would be great to include the context of how much runtime a numerical solver needs to simulate Navier-Stokes. Whichever outcome (in terms of "who's faster") is fine, but the context would help assess the efficiency of the active-learning methods. If the solver is slow, the context would underline the statement in line 034 that states how costly numerical solvers can be.
>
> We agree this would strengthen our motivation. Although the numerical solver for Navier-Stokes equation only takes a few tens of seconds per round, the numerical solver for KdV equation without batch processing takes 1,232 seconds per round, which could be effective in underlining our statement about how costly numerical solvers can be. We will add this information to Section 5.8.
>
> We would like to thank you again for the valuable feedbacks and questions. We will provide answers to the "less important" questions soon. In the meantime, please feel free to ask us if you have any further questions or comments.

---

> > ### Comment · Reviewer_NgwX · 2024-11-15
> >
> > Thank you for the comprehensive reply. I'm looking forward to seeing the promised changes in the manuscript.
> >
> > Thank you also for clarifying the FlexAL 10 is not a cut in the number of training epochs. I must've misunderstood this part in my initial review.
> >
> > >  in the greedy selection algorithm. Table 14 in the Appendix shows that the performance of FlexAL 10 is comparable to that of FlexAL, and always outperforms the best baseline method
> >
> > Table 14 shows this in a single experiment, and all other studies involve FlexAL instead of FlexAL 10. If FlexAL 10 is faster than FlexAL (which it seems to be), and if its output is comparable to FlexAL (which Table 14 suggests), I am now wondering why FlexAL 10 is not the default method for all other experiments. But I suppose that the choice of $T$ depends on the problem.
> >
> > Thanks again for all the clarifications. I have increased my score, assuming that the promised revisions will end up in the manuscript.

---

> ### Author Response · Authors · 2024-11-15
>
> Thank you increasing the score. We will update you with the revised manuscript as soon as possible.
>
> To answer your question about why FlexAL 10 is not the default method, we would like to note that FlexAL 10's performance in all four equations as reported in Table 14 lies almost in the middle of baseline and FlexAL. It thus might not have been appropriate to call FlexAL 10 exactly "comparable" to FlexAL, but it is definitely superior to all the baseline methods.
>
> Below are answers to the "less important" questions. We thank you again for taking the time to provide such detailed and valuable feedbacks.
>
> > Line 053, "we argue that querying all the states (...) is not cost-efficient": does this sentence perhaps require more nuances? While sparse subsets of trajectories decrease the complexity (on paper), the suggested procedure for finding them is sufficiently expensive that the proposed algorithm is more costly than full-batch versions (Section 3.3, Table 4).
>
> We will clarify in the paper that what we mean by "cost-efficient" is that we reduce the cost of data acquisition required for achieving a certain accuracy. A better wording might have been "sample-efficient".
>
> > Line 095: Perhaps Brandstetter et al. (2022b) are not the best reference for spatiotemporal PDEs. Personally, I don't think this sentence needs any reference, but if one should be used, perhaps something like Evans' "Partial Differential Equations" book would be more appropriate.
>
> Thank you for the suggestion, we agree that it probably doesn't need a reference. We will make this revision.
>
> > Line 100: This statement about existence would benefit from a reference.
>
> Thank you for the suggestion. Our claim should actually be fixed as follows: "If the PDE and boundary conditions are *well-posed*, then there exists a unique evolution operator ...". We will add reference to the fact that all the PDEs used in our experiments are well-posed.
>
> > Line 112: What does "primary focus" mean here? It seems to be the only focus. Have I missed something?
>
> No, you haven't missed anything. We will rephrase it as "In this
> paper, we focus on the autoregressive trajectory prediction task".
>
> > Line 128: Why does this sentence introduce a distribution of initial conditions, but the rest of the manuscript (for example, Algorithm 1) operates on a pool of initial conditions? I understand that the pool is sampled from the condition, but it might be more reader-friendly to use a pool of conditions throughout.
>
> This sentence was intended to acknowledge that the test data are sampled from the same distribution as the pool used for training. We agree that this might be confusing, and will instead rephrase it as "We assume that there exists a pool $P$ of initial conditions." We will make the comment about the test data in the experiments sections.
>
> > Line 262: How do the results depend on this choice of T and epsilon?
>
> We haven't done extensive experiments with different values of $T$ and $\epsilon$. Increasing $T$ improves performance until it plateaus. Increasing $\epsilon$ values higher than a certain point deteriorates the performance slighly. On the other hand, decreasing $\epsilon$ too much also deteriorates the performance, but can be recovered with a higher value of $T$, leading to higher computational cost. We will add these in the revision.
>
> > Line 298: The abbreviation "QbC" is used early in the paper (for instance, in line 117 or 204). Maybe it would be good to introduce it before line 298.
>
> Thank you for pointing this out. We will introduce it earlier in line 117.

---

> > ### Comment · Reviewer_NgwX · 2024-11-15
> >
> > Thanks for these explanations!

---

### Official Review · Reviewer_rtnG · 2024-10-31

**Soundness:** 3
**Presentation:** 3
**Contribution:** 3
**Rating:** 5
**Confidence:** 3

**Summary:**

The authors propose a new approach of data acquisition for active learning. They suggest considering a subset of the time steps from a standard numerical solver along a trajectory, and use a fitted surrogate model to approximate the remaining of data. Given the proposed strategy does not provide a significant improvement compared to alternative methods consistently, I believe the current version of manuscript should not be accepted.

**Strengths:**

The paper makes a nice comparison between their suggested method of active learning and the other available methods in the literature.

**Weaknesses:**

I think the presented idea lacks novelty, and the shown numerical results are not suggesting any significant improvement compared to other methods.

**Questions:**

**Major:**

- The proposed method does not seem to have a significant improvement compared to the other compared methods. For example in Fig3, RMSE of proposed strategy is slightly (around 10%) better than SBAL, random, or QbC, for KdV, KS, and NS. I wonder why the method performs so much better in case of heat equation. It definitely needs further investigation.

**Minor:**

- P2 line95, What is the space of $\mathbb{X}$? Please define the space.
- P2 line 99, I highly doubt uniqueness of the operator G_{t0}. Given solution at $t=0$ and $t=Delta t$, there may exist infinite PDEs that satisfy both initial and final condition.
- P3 eq. 2, Wouldn't PINN loss help here, if the PDE is known?
- Algorithm 1, line 7: please clarify notation.

---

> ### Author Response · Authors · 2024-11-15
>
> We appreciate your thoughtful questions and feedbacks. We provide a simple explanation of our method in **Common Response 1. Simple explanation** in hopes of clarifying some confusions.
>
> > The proposed method does not seem to have a significant improvement compared to the other compared methods. For example in Fig3, RMSE of proposed strategy is slightly (around 10%) better than SBAL, random, or QbC, for KdV, KS, and NS. I wonder why the method performs so much better in case of heat equation. It definitely needs further investigation.
>
> **We cannot see any ambiguity in the significance of our results.** Musekamp et al. [1] provide experiments with baseline methods on PDEs, and report marginal improvement over Random on challenging equations such as KS and NS. This is consistent with our results: for KS and NS, there is hardly any improvement with existing baseline methods (QbC, SBAL, LCMD) over Random, and some methods even underperform compared to Random. FlexAL is the only method that consistently improves over Random, and it does so by a significant margin. In the table below, we report the improvement in average log RMSE from Random, and the ratio between the improvement of FlexAL and that of the best baseline method. FlexAL provides two to six times improvements over the best methods.
>
> Table 1. Average log RMSE, improvement over Random.
>
> | Equation | QbC | LCMD | SBAL | SBAL+FlexAL | Ratio (SBAL+FlexAL / best) |
> |----------|-----|------|------|------------|------------------------|
> | Heat | -0.236 | -0.053 | -0.213 | **-0.616** | 2.6 |
> | KdV | 0.075 | 0.065 | -0.161 | **-0.279** | 1.7 |
> | KS | -0.010 | 0.304 | -0.017 | **-0.091** | 5.4 |
> | NS | -0.007 | 0.032 | -0.002 | **-0.042** | 6.0 |
>
> We encourage you to look at the results in Figure 4 of Musekamp et al. [1], Figure 3(b) of Li et al. [2] (the yellow and blue lines correspond to Random and Coreset), and Figure 5 of Bajracharya et al. [3]. Our method provides arguably the largest and the most robust performance gain reported in the PDE active learning literature. In Appendix B, we have reported other metrics such as normalized RMSE, MAE, and 99%, 95%, 50% quantiles of RMSE, which all point to the superiority of FlexAL over baselines. Our results also show strong statistical signficance as evidenced by the non-overlapping error bars.
>
> Most importantly, we should consider the *data efficiency*, that is, the amount of data required to attain a certain accuracy. Consider the NS experiment for example. The RMSE of SBAL+FlexAL at the fifth round is lower than the RMSE of the best baseline at the tenth round. FlexAL therefore halves the amount of data required to attain this accuracy. Carrying out this analysis on all equations, we see reductions in data cost ranging from 10% to 50%. A 10% reduction in data acquisition cost is far from a minor improvement. This can translate to hours, days, or even weeks of time saved by FlexAL in real world simulations.
>
> All in all, the **robust** and **large** performance gains of FlexAL are unprecedented in the PDE active learning literature, and we believe this is a significant contribution to the field.
>
> > I think the presented idea lacks novelty.
>
> We explore, for the first time, the idea of simulating PDE trajectories with a mixture of numerical solvers and surrogate models. We also propose an active learning method that adaptively chooses which time steps to query to the numerical solver or the surrogate model. The proposed acquisition function is novel and also generally applicable, which we believe is a fundamental contribution to the field of active learning. We suggest reading **Common Response 1. Simple explanation** for a simplified explanation of our method, which we hope gives a clearer picture of our method's novelty.
>
> References:
>
> [1] Daniel Musekamp, Marimuthu Kalimuthu, David Holzmüller, Makoto Takamoto, and Mathias Niepert. Active learning for neural pde solvers. *arXiv preprint arXiv:2408.01536*, 2024.\
> [2] Shibo Li, Xin Yu, Wei Xing, Robert Kirby, Akil Narayan, and Shandian Zhe. Multi-resolution active learning of fourier neural operators. In *International Conference on Artificial Intelligence and Statistics*, pp. 2440–2448. PMLR, 2024.\
> [3] Pradeep Bajracharya, Javier Quetzalcóatl Toledo-Marín, Geoffrey Fox, Shantenu Jha, and Linwei Wang. Feasibility study on active learning of smart surrogates for scientific simulations. *arXiv preprint arXiv:2407.07674*, 2024.

---

> ### Author Response · Authors · 2024-11-15
>
> > P2 line95, What is the space of \mathbb{X}? Please define the space.
>
> $\mathbb{X}$ is a general notation for any spatial domain. In our case, it would be $\mathbb{X}=I$ where $I=[0,1]$, and for 2D Navier-Stokes, $\mathbb{X} = I^2$.
>
> > P2 line 99, I highly doubt uniqueness of the operator G_{t0}. Given solution at $t=0$ and $t=Deltat$, there may exist infinite PDEs that satisfy both initial and final condition.
>
> The PDE defines the operator $G_{t_0}$, not the other way around. For example, if we have a wave equation, $G_{t_0}$ takes the wave shape at time $t_0$ and outputs its shape at $t_0 + \Delta t$. P2 line 99 is saying that this operator $G_{t_0}$ is unique to each PDE.
>
> > P3 eq. 2, Wouldn't PINN loss help here, if the PDE is known?
>
> Using the residual loss as in PINN can help, and there are indeed active learning methods for PDEs that use this loss as an acquisition function. However, a low residual loss at a point doesn’t necessarily mean that the model has low error globally, since the error can accumulate at earlier time steps. Moreover, it can’t be applied to a more general setting where the PDE is unknown. We kindly refer you to Section 4.1 for a survey of the existing AL methods in PDEs, including ones that use the residual loss.
>
> > Algorithm 1, line 7: please clarify notation.
>
> The $\oplus$ symbol stands for XOR operation. Algorithm 1, lines 6 and 7 describe the process of applying random bit-flips to $S$ to obtain $S'$.
>
> We will make sure to add these clarifications in a revised manuscript. Thank you for asking these, and please feel free to ask any other questions that you have.

---

> > ### Comment · Reviewer_rtnG · 2024-11-25
> >
> > I thank the authors for their clarifications. Still, there are some details missing in the manucript.
> >
> > > The PDE defines the operator $G_{t_0}$, not the other way around. For example, if we have a wave equation, $G_{t_0}$ takes the wave shape at time $t_0$ and outputs its shape at $t_0 + \Delta t$. P2 line 99 is saying that this operator $G_{t_0}$ is unique to each PDE.
> >
> > Is $G$ related to the Green function of PDE? I am still not convinced why G is unique for a well-posed PDE. The authors cite Evan's book, but do not explain this issue or make a proper reference.
> >
> > On a separate note, if the PDE is solved on a grid, then why the term "trajectory" is used for the proposed method? Trajectory implies some sort of particle method, which is not the case here. My understanding of your method is that the numerical method is used to fill out the data set when there is missing data at some times.

---

> > > ### Comment · Reviewer_WEGs · 2024-11-25
> > >
> > > FWIW I tend to support the authors' terminology regarding "trajectory" being acceptable; if the path of that trajectory happens to be function-valued at each time step, I think it is still a trajectory.

---

> > > > ### Comment · Reviewer_WEGs · 2024-11-25
> > > >
> > > > >I am still not convinced why G is unique for a well-posed PDE
> > > >
> > > > This is my understanding of the definition of _well-posed_ --- that the forward evolution of the system is unique (and ill-posed admits multiple solutions). I believe that saying the _evolution operator itself_ is unique is slightly more involved, because it would involved some technical conditions on the function spaces that the evolution operator maps between; is this what reviewer WEGs is concerned about? If so, can we simply delete the word "unique"? I'm not sure the uniqueness  is actually used in the paper (?); so long as we recover something with an action that is close in some sense we are happy, as far as I can tell.

---

> ### Author Response · Authors · 2024-11-24
>
> We thank you again for the insightful questions and feedbacks. We have posted a response to your concerns and a revised version of our manuscript. As we reach the end of our discussion period, your response would be of great help in further improving our work.

---

> ### Author Response · Authors · 2024-11-25
>
> Thank you, reviewer rtnG, for asking these question, and thank you, reviewer WEGs, for kindly sharing your understanding.
>
> > Is $G$ related to the Green function of PDE? I am still not convinced why G is unique for a well-posed PDE. The authors cite Evan's book, but do not explain this issue or make a proper reference.
>
> As given in page 7 of Evan's book, a PDE is *well-posed* if
> - the problem has a solution
> - this solution is unique
> - the solution depends continuously on the data given in the problem
>
> That is, given an initial condition at $t_0$, we have a unique solution at $t_0 + \Delta t$. However, as reviewer WEGs has kindly pointed out, the uniqueness implied by well-posedness requires some mathematical care when taken literally. We will therefore remove the word "unique" in the next revised version, since the uniqueness is not used in our paper: we are simply training a surrogate model $\hat G$ given an evolution operator $G$.
>
> > On a separate note, if the PDE is solved on a grid, then why the term "trajectory" is used for the proposed method? Trajectory implies some sort of particle method, which is not the case here. My understanding of your method is that the numerical method is used to fill out the data set when there is missing data at some times.
>
> The "trajectory" that we refer to is the trajectory of PDE states $u(t, \cdot)$ over time $t$. This has been used, for example, in [1]. P2 line 104 also clarifies its definition. As reviewer WEGs has pointed out, we are referring to a trajectory of functions (PDE states) over time.
>
> [1] Lippe, Phillip, et al. "Pde-refiner: Achieving accurate long rollouts with neural pde solvers." Advances in Neural Information Processing Systems 36 (2024).

---

> > ### Comment · Reviewer_rtnG · 2024-11-25
> >
> > As mentioned, a well-posed PDE has a unique solution. This does not mean that there exists a unique map G that maps solution from $t_0$ to $t_0+Delta t$. A solution to a PDE, could be solution to other PDEs as well.
> >
> > I don't agree with the terminology used here. Then, solution of any PDE with a temporal (time dependent) term should be called trajectory. To me, sounds like rebranding a known term, even if a NeurIPS paper uses it.
> >
> > I increased the score.

---

> ### Author Response · Authors · 2024-11-25
>
> > As mentioned, a well-posed PDE has a unique solution. This does not mean that there exists a unique map G that maps solution from $t_0$ to $t_0+Delta t$. A solution to a PDE, could be solution to other PDEs as well.
>
> We think you might have been misled to think that we are talking about specific states at $t_0$ and $t_0 + \Delta t$. This is not the case. We are suggesting that there is a PDE-specific operator that, given *any* state at $t_0$, outputs the PDE-specific evolved state at $t_0 + \Delta t$. If this still doesn't answer your question, please do not hesitate to ask further.
>
> > I don't agree with the terminology used here. Then, solution of any PDE with a temporal (time dependent) term should be called trajectory. To me, sounds like rebranding a known term, even if a NeurIPS paper uses it.
>
> That is exactly how we define "trajectory" in our paper. It is the solution of any PDE with a temporal (time dependent) term. We find that this is an intuitive term to use in our context, and would be happy to hear your suggestion for new terms.
>
> We would also appreciate if you could share any other concerns or questions about our paper, other than the above two.

---

> > ### Comment · Reviewer_rtnG · 2024-11-25
> >
> > Consider the wave equation u=sin(x-t). This is a solution to infinite PDEs, for example ut+ux=0 as well as  ut+ux+uxx+uxxxx=0 among others. Each PDE in essence is acting as a map, bringing solution from t0 to t0+delta t. Since you are learning the map from data, note that for a given solution (data), the map is not unique. So, unlike what you are suggesting, I think what you call function/operator/machine is not unique.
> >
> > > We find that this is an intuitive term to use in our context
> >
> > I disagree. In numerical methods for PDEs, trajectories are used for Lagrangian or particle based methods.

---

> ### Author Response · Authors · 2024-11-25
>
> The ground truth operator $G$ is not something learnt from data. It is defined given a PDE, not the data from the PDE. We are learning a surrogate of $G$, namely $\hat G$, from some data. There are, of course, infinitely many $\hat G$ that perfectly fit to the data. But there is only one unique $G$ for a PDE.
>
> Regarding the use of the term "trajectory", do you think replacing it with "solution" would then be more appropriate? We would be happy to hear your suggestion.

---

### Official Review · Reviewer_PXq7 · 2024-11-03

**Soundness:** 2
**Presentation:** 3
**Contribution:** 3
**Rating:** 8
**Confidence:** 4

**Summary:**

The authors propose an active learning framework for training surrogate models that can aid in solving partial differential equations. Traditional solvers for partial differential equations are computationally expensive, which motivated the development of surrogate models to efficiently solve PDEs. However, for training these surrogate models, costly numerical simulation data are required. Current active learning based strategies for training such surrogate models require entire PDE trajectories from a given starting condition, which is costly. The authors propose a flexible sampling strategy, which does not require entire PDE trajectories, and only samples upto a given budget. Experiments show that the method performs better in terms of RMSE compared to baseline.

**Strengths:**

The motivation of the work is presented well. The related literature is reviewed well in the introduction section and related work section.

**Weaknesses:**

The methodology of the paper requires a lot more elaboration. Here are a few points that are not clearly answered in the current manuscript:
How does sparse sampling works for numeric solvers? For example, given a pattern S = {T, F,F,..., T}, which is basically sampling an initial condition and final condition, how will we sample the data here without sampling all intermediate states? If we cannot do that, then how does the method reduce the cost of acquiring training data?
How much computation cost, data acquisition cost is reduced in this framework? The experimental results report the accuracy of the trained model, however since the original motivation of the work is about reducing such costs, presenting these additional statistics makes more sense than only reporting model accuracy alone.
Algorithm 1 should be described line by line, probably best to do this at the end of section 3.3
Figure 1 needs some rethinking, currently it is difficult to see the author’s motivation, or the entire framework from this figure alone. Perhaps show how the cost increases with added data acquisition side by side with the baseline and FLEXAL strategy

**Questions:**

See weakness section above

---

> ### Author Response · Authors · 2024-11-15
>
> We appreciate your thoughtful questions and feedbacks. We provide a simple explanation of our method in **Common Response 1. Simple explanation** in hopes of clarifying some confusions.
>
> > For example, given a pattern S = {T, F,F,..., T}, which is basically sampling an initial condition and final condition, how will we sample the data here without sampling all intermediate states? If we cannot do that, then how does the method reduce the cost of acquiring training data?
>
> As described in Section 3.1, the intermediate states are sampled with the surrogate model $\hat G$.
>
> > How much computation cost, data acquisition cost is reduced in this framework? The experimental results report the accuracy of the trained model, however since the original motivation of the work is about reducing such costs, presenting these additional statistics makes more sense than only reporting model accuracy alone.
>
> As described in Section 2.2, each round of AL incurs the same data acquisition cost of $B$. Therefore, the graphs of the RMSE with respect to the AL round, such as Figure 3, can also be seen as graphs of RMSE with respect to the cost. We are effectively reducing the cost required to attain the same accuracy.
>
> We can see how this can be confusing for the readers. We intend to clarify this point by explaining it in the experimental results section. Thank you for the question.
>
> > Algorithm 1 should be described line by line, probably best to do this at the end of section 3.3.
>
> We appreciate this feedback. Although the last paragraph of Section 3.3 is intended to be a description of Algorithm 1, we failed to explain some notations and steps in Algorithm 1. We therefore plan modify the explanations as below. Modifications are highlighted in boldface.
>
> “Specifically, a full-trajectory AL method **\mathcal A**, which we call a \textit{base} method, first selects an initial condition $u^0$.”
>
> “In the greedy algorithm, we start by initializing $S$ with all entries set to \mathsf{true}. At each step, we propose a neighboring pattern **S’** by applying a bit-flip mutation, where each bit of $S$ is flipped with a probability of $\epsilon$. The proposal is accepted only if the acquisition value **a^\*(\bm u^0,S')** is higher than the current value **a^\*(\bm u^0, S)**. **This process of proposal and acceptance/rejection is repeated for T times.**"
>
> > Figure 1 needs some rethinking, currently it is difficult to see the author’s motivation, or the entire framework from this figure alone. Perhaps show how the cost increases with added data acquisition side by side with the baseline and FLEXAL strategy
>
> We appreciate this feedback. We would first like to ask if the motivation behind Figure 1 might have become clearer to you after reading our answers above. In addition, we have modified the figure such that the number of queries to the numerical solver (black solid lines) is equal for both sides, better reflecting our fixed budget framework.
>
> https://anonymous.4open.science/api/repo/iclr2025rebuttal-18AB/file/Figure1_v2.png?v=189a6665
>
> We also plan to add a more detailed explanation of Figure 1 in Section 3.1 as follows:
>
> "In other words, the numerical solver and surrogate model are used selectively to acquire each time step. **Fig. 1 summarizes our framework. Each dot represents a PDE state, and a path connecting two dots represents a time step of a simulation. The black solid lines are obtained with numerical solvers while the red dotted lines with surrogate models. Both sides have the same number of black solid lines, reflecting the fact that we sample with a fixed budget $B$ at each round. Our method, FlexAL, can sample many more trajectories under the same budget.**"
>
> If you think Figure 1 needs other modifications, we would be more than happy to hear your thoughts.

---

> ### Author Response · Authors · 2024-11-24
>
> We thank you again for the insightful questions and feedbacks. We have posted a response to your concerns and a revised version of our manuscript. As we reach the end of our discussion period, your response would be of great help in further improving our work.

---

> > ### Comment · Reviewer_PXq7 · 2024-11-26
> >
> > I sincerely thank the authors for their detailed explanation. I think my understanding of the paper is much better now after the revisions proposed by the authors. Regarding figure 1, indeed the detailed explanation now makes the motivation and authors' contribution clearer, I would suggest to put this into the caption if not done already. I have increased my review score, I wish the authors very best for their hard work.

---

> ### Author Response · Authors · 2024-11-26
>
> Thank you for the positive review. We have added the explanation to the caption. We thank you again for all the constructive feedbacks.

---

### Author Response · Authors · 2024-11-15

**Common Response 1. Simple explanation**

We provide a simplified explanation of our method.

Given a PDE, a time-interval $\Delta t$, and a numerical solver $G$ that computes how a PDE state evolves over $\Delta t$, we aim to train a surrogate model $\hat G$ using data from $G$. For example, if we have a wave equation, $G$ takes the current wave shape and computes its shape after $\Delta t$ seconds. Since running $G$ is expensive, we want to acquire training data efficiently using active learning.

Unlike existing methods that acquire entire trajectories using $G$, we propose acquiring trajectories using a mixture of $G$ and our current surrogate $\hat G$. The process:

1. Select an initial condition $u^0$
2. Choose a sampling pattern $S=(b_1, \dots, b_L)$ (e.g., $S = (\mathsf{true}, \mathsf{false}, \dots, \mathsf{true})$)
3. Simulate the $i$ th time step using $G$ when $b_i$ is $\mathsf{true}$, $\hat G$ otherwise
4. Add only $G$'s input-output pairs to training data

is repeated until the batch reaches a fixed budget $B$. With budget $B = 32$:
- Traditional: 2 full trajectories of length 16
- FlexAL: ~8 trajectories with 4 key time steps each


We propose a novel acquisition function to choose optimal patterns $S$. We emphasize again that we never train on $\hat G$'s outputs — it only bridges gaps between solver-generated states.


To summarize, our method allows for sampling of more trajectories under the same budget, compared to traditional methods which are special cases of ours where $S = (\mathsf{true}, \mathsf{true}, \dots, \mathsf{true})$. Our method benefits the surrogate model by improving the diversity of the training set.

---

### Author Response · Authors · 2024-11-20

**Common Response 2. Summary of revision to our paper**

We have posted a revision to our paper with all changes highlighted in red. We have aimed to address the concerns about clarity, motivation, and technical details. Key changes include:

Clearer Explanation of Method and Framework:
- Added a simplified explanation in Section 3.1 of how our method combines numerical solvers and surrogate models to acquire partial trajectories
- Clarified the aim and motivation of our paper in Section 2.2
- Revised Figure 1 to better illustrate the fixed budget framework, showing equal numbers of numerical solver queries (black lines) on both sides
- Moved the complex Batch Acquisition Algorithm details to the appendix and added a new high-level algorithm showing the complete framework including surrogate model training

Technical Clarifications:
- Expanded discussion of computational complexity and runtime considerations in Section 5.7
- Added detailed results showing wall-clock times across all equations in Table 15
- Added Appendix D.1 explaining the motivation behind our acquisition function design, including discussion of alternative approaches we considered
- Added normalized RMSE results in Appendix B to provide scale-independent performance metrics
- And many other small details

We believe these revisions make the paper clearer while better highlighting our key contributions. We welcome any additional feedback from the reviewers.

---

### Comment · Area_Chair_7hnb · 2024-11-25
**Reviewers' Response**

Dear Reviewers,

As the author-reviewer discussion period is approaching its end, I would strongly encourage you to read the authors' responses and acknowledge them, while also checking if your questions/concerns have been appropriately addressed.

This is a crucial step, as it ensures that both reviewers and authors are on the same page, and it also helps us to put your recommendation in perspective.

Thank you again for your time and expertise.

Best,

AC

---

### Meta-Review · Area_Chair_7hnb · 2024-12-20

**Metareview:**

The paper proposes an active learning framework for efficiently training surrogate models to solve PDEs. The authors introduce a "sparse" sampling strategy, in contrast to many existing active learning methods for surrogate models that rely on full simulation trajectories. This strategy involves selectively sampling a subset of time steps from a numerical solver along a trajectory, guided by the disagreement among an ensemble of surrogate models, and using a trained surrogate to approximate the remaining steps. The authors benchmark their idea on a set of three one-dimensional and one two-dimensional equations.

Reviewers are skeptical of whether it offers significant improvement over existing methods, particularly given the limited empirical evidence provided and the small gains reported. The core idea of using model disagreement to guide sample selection is acknowledged, but concerns remain regarding the method's overall effectiveness and the strength of its empirical validation on typical PDE benchmark problems. In particular, the parameters of the chosen PDEs result in overly smooth solutions that do not propagate much (even for the KdV equation), and as such, it is unclear how the framework would actually work in the case of chaotic systems or systems with rapid transitions. Therefore, the impact of the paper for PDE emulators is quite limited. In fact, using solutions that exhibit shocks, highly dispersive waves, or highly chaotic behavior (such as Navier-Stokes with a much higher Reynolds number) would make this paper much stronger.

Given the lack of convincing numerical results to back up the claims, I recommend rejection.

**Additional Comments On Reviewer Discussion:**

The authors updated several points raised by the reviewers, although, some of the main issues (the selection of the PDEs and their very smooth solutions remains).

---

### Decision · Program_Chairs · 2025-01-22

Reject